# Spread of pathological tau proteins through communicating neurons in human Alzheimer's disease

Jacob W. Vogel[1✉], Yasser Iturria-Medina [1], Olof T. Strandberg[2], Ruben Smith [2,3], Elizabeth Levitis[1], Alan C. Evans[1,92], Oskar Hansson [2,3,92✉], Alzheimer's Disease Neuroimaging Initiative* & the Swedish BioFinder Study*

Tau is a hallmark pathology of Alzheimer's disease, and animal models have suggested that tau spreads from cell to cell through neuronal connections, facilitated by $\beta$-amyloid (A$\beta$). We test this hypothesis in humans using an epidemic spreading model (ESM) to simulate tau spread, and compare these simulations to observed patterns measured using tau-PET in 312 individuals along Alzheimer's disease continuum. Up to 70% of the variance in the overall spatial pattern of tau can be explained by our model. Surprisingly, the ESM predicts the spatial patterns of tau irrespective of whether brain A$\beta$ is present, but regions with greater A$\beta$ burden show greater tau than predicted by connectivity patterns, suggesting a role of A$\beta$ in accelerating tau spread. Altogether, our results provide evidence in humans that tau spreads through neuronal communication pathways even in normal aging, and that this process is accelerated by the presence of brain A$\beta$.

[1] Montreal Neurological Institute, McGill University, Montréal, QC, Canada. [2] Clinical Memory Research Unit, Lund University, Lund, Sweden. [3] Memory Clinic, Skåne University Hospital, Lund, Sweden. [92] These authors contributed equally: Alan C. Evans, Oskar Hansson. *Lists of authors and their affiliations appear at the end of the paper. ✉email: jacob.vogel@mail.mcgill.ca; oskar.hansson@med.lu.se

Alzheimer's disease is characterized by the presence of β-amyloid plaques and neurofibrillary tangles of hyper-phospohrylated tau at autopsy. Both of these pathological phenomena can now be quantified spatially in the brains of living humans using positron emission tomography (PET), allowing for the study of disease progression before death and, indeed, before symptoms manifest[1]. β-Amyloid plaques are detectable in the brain many years or even decades before dementia onset[2], but appear to have only subtle effects on cognition and brain health in humans[3–6]. In contrast, tau neurofibrillary tangles are strongly correlated with local neurodegeneration and, in turn, cognitive impairment[7,8]. However, tau tangle aggregation specifically in the medial temporal lobes is a common feature of normative aging[9–11], itself associated with subtle cognitive effects[12,13]. Frank cognitive impairment often coincides with the spreading of tau tangles out of the medial temporal lobes and into the surrounding isocortex, a process that animal models have suggested may be potentiated or accelerated by the presence of β-amyloid plaques[14,15].

Due to its close link with neurodegeneration and cognitive impairment, tau has received special attention as a potential therapeutic target for Alzheimer's disease[16]. Perhaps the most compelling features of tau pathophysiology are its rather focal distribution of aggregation and its highly stereotyped pattern of progression through the brain. Specifically, neurofibrillary tangles first appear in the transentorhinal cortex, before spreading to the anterior hippocampus, followed by adjacent limbic and temporal cortex, association isocortex, and finally to primary sensory cortex[10,17–19]. This very particular pattern has led many to speculate that pathological tau itself, or a pathological process that incurs tau hyper-phosphorylation and toxicity, may spread directly from cell to cell through anatomical connections[20,21]. Strong evidence in support of this hypothesis has come from animal models, which have repeatedly demonstrated that human tau injected into the brains of β-amyloid-expressing transgenic rodents leads to the aggregation of tau in brain regions anatomically connected to the injection site[14,22–25]. An important caveat to the aforementioned studies is that they often involve injection of tau aggregates that greatly exceed the amount of tau produced naturally in the human brain. In addition, the studies were performed in animals that do not get Alzheimer's disease naturally.

Unfortunately, there are many obstacles to studying the tau-spreading hypothesis in humans. While autopsy studies have provided evidence for tau spreading[26,27], this evidence comes in the form of limited snapshots in deceased individuals. Tau-PET allows for the quantification of tau in vivo, but the PET signal is contaminated by off-target binding that limit interpretations[28–32]. Despite this limitation, circumstantial evidence has emerged supporting the hypothesis that tau spreads through connected neurons in humans. Studies decomposing the spatial distribution of tau-PET signal in the human brain have revealed spatial patterns highly reminiscent of brain functional networks[33–35]. In addition, brain regions with greater functional connections to the rest of the brain tend to have greater tau accumulation[36], regional connectivity is associated with longitudinal changes in tau burden[37], and correlations have been found between functional connectivity patterns and tau covariance patterns[38,39].

Despite mounting evidence linking brain connectivity and tau expression, the aforementioned studies mostly either involve comparisons between coarse whole-brain measures of tau and brain connectivity, or are limited to only a fraction of brain connections. The initial seeding of tau in the cortex is thought to lead subsequently to secondary seeding events that cascade systematically through the cerebral cortex. Therefore, it is paramount that studies assessing the spread of tau through the brain can effectively model the complex spatio-temporal dynamics of this process. Therefore, we test the tau-spreading hypothesis by placing a "tau seed" in the entorhinal cortex, simulating its diffusion through measured functional and anatomical connections, and comparing the simulated pattern of global tau spread with the actual pattern derived from tau-PET scans of 312 individuals. This method allows for a cascade of secondary tau seeding events to occur along a network over time, more closely simulating proposed models of tau spread in the brain. We then examine how the behavior of our model interacts with brain β-amyloid and what it can tell us about asymmetric tau distribution.

## Results

**Sample information.** Flortaucipir (AV1451)-PET scans measuring tau neurofibrillary tangles in vivo were available for 312 individuals spanning the Alzheimer's disease spectrum. Demographic information for this sample can be found in Table 1.

**Tau-positive probabilities enhance fidelity of tau-PET data.** We executed a procedure to mitigate off-target binding of Flortaucipir-PET data using mixture modeling. Regional Gaussian mixture modeling of Flortaucipir SUVR data across all subjects suggested a two-component (bimodal) model as a superior fit for all 66 cortical regions of interest, including the left and right hippocampi and amygdalae. These 66 regions were converted to tau-positive probabilities (Fig. 1c) using the Gaussian mixture models. This threshold-free, data-driven transformation yielded a sparse data matrix with a clear pattern suggesting a gradual progression of tau across regions of the brain (Supplementary Fig. 1). When sorted from least to most tau (e.g. ref. [18]), the regional ordering greatly resembled the previously described progression of tau pathology[17] (Fig. 2).

**Neuronal connectivity explains the spatial pattern of tau.** An epidemic spreading model (ESM) was fit to the data, simulating the spread of tau from a single epicenter through macroscale brain connections over time (Fig. 1). The ESM was fit over several regional tau-PET datasets resulting from combinations of arbitrary data pre-processing decisions (see Methods). All models were fit using the left and right entorhinal cortex as the model epicenter. Models performed best when SUVR data for the 66 cortical regions were converted to tau-positive probabilities as described above, with regression of age, sex, and non-specific choroid plexus binding from the data occurring beforehand (Supplementary Fig. 2A, B, F). Partial volume correction (PVC) (Supplementary Fig. 2C) and exclusion of Aβ− MCI individuals (Supplementary Fig. 2E) did not appear to impact model performance, though the best-fitting model did not use PVC and excluded Aβ− MCI individuals (Supplementary Fig. 2A).

The best-fitting model was fit over a system of anatomical connections created from a separate sample of young, healthy individuals using diffusion tensor imaging (DTI) tractography. This model explained 70.2% (null model mean $r^2$ [95% CI] =

**Table 1 Demographic information.**

|  | CN | MCI | AD | Total |
|---|---|---|---|---|
| $n$ | 162 | 89 | 61 | 312 |
| Age (SD) | 72.0 (6.4) | 70.84 (7.8) | 72.0 (7.9) | 71.7 (7.1) |
| % Women | 45.1 | 64.0 | 58.6 | 53.1 |
| Education (SD) | 14.8 (3.6) | 15.3 (3.7) | 12.8 (3.9) | 14.6 (3.8) |
| %APOE4 | 41.9 | 58.4 | 68.5 | 51.7 |
| % amyloid positive | 42.6 | 64.0 | 100.0 | 66.2 |

CN cognitively normal, MCI mild cognitive impairment, AD Alzheimer's disease dementia, SD standard deviation.

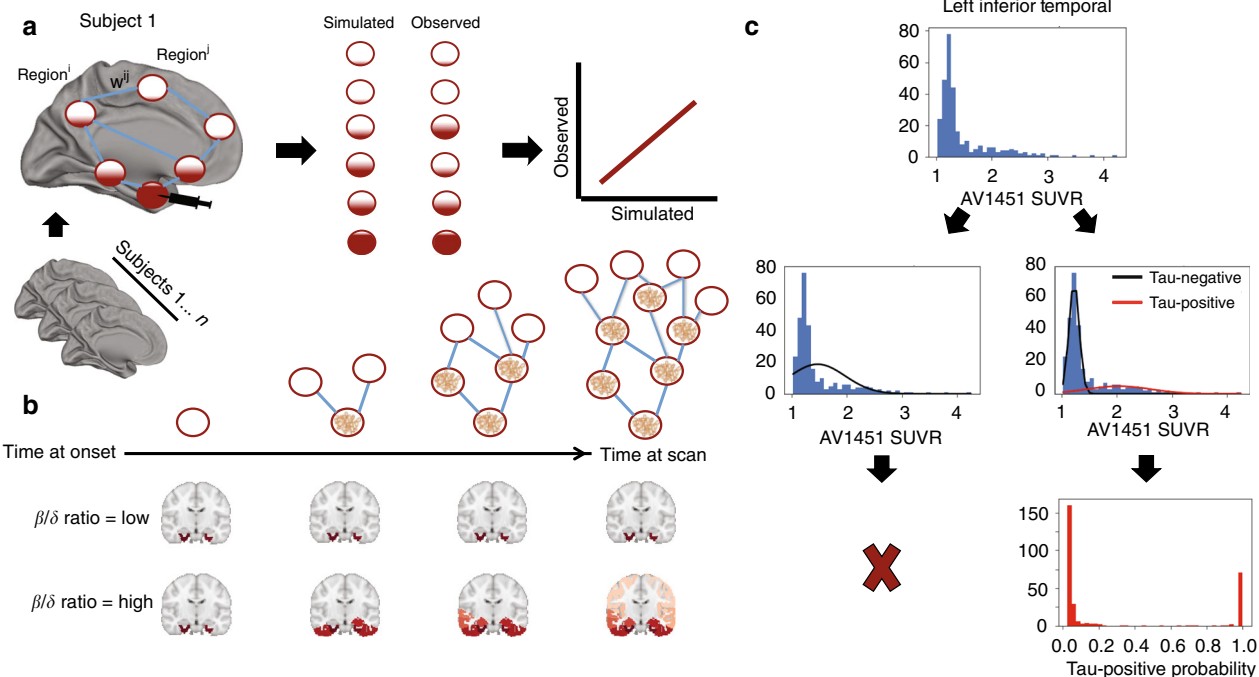

**Fig. 1 Methodological approaches. a** An artificial system based on a pairwise relationship (e.g. functional connectivity) matrix is created, where the relationship between regions *i* and *j* is represented by weight *ij*. For each subject, a seed is placed at the model epicenter, and the diffusion of this signal over time is simulated through the system, where the inter-regional relationships determine the pattern of spread, and subject-level free parameters determine the velocity of diffusion, until an optimal fit is reached. The simulated tau signal is then compared to the observed tau-PET signal to evaluate the model. **b** Advantages of the ESM over traditional approaches includes the initiation of secondary seeding events as the diffusion process reaches new regions (top), and the fitting of subject-level production (*β*) and clearance (*δ*) parameters. A balance in these parameters will lead to little to no spreading over time, while increasing imbalance leads to accelerated spread. **c** The distribution of all SUVR values in the left inferior temporal ROI are shown. Two Gaussian mixture models are fit to the data. When a one-component model fits the data better, the ROI is discarded. When a two-component model fits better, the probability that each values falls upon the second distribution is calculated.

0.056 [0.016, 0.135], $p < 0.01$) of the overall spatial pattern of tau (Fig. 3a), and on average, explained 50.9% (SD = 21.8%; null model mean $r^2$ [95% CI] = 0.104 [0.077, 0.147], $p < 0.01$) of the spatial pattern within individual subjects (Fig. 3a). Importantly, across all possible regions of interest, the entorhinal cortex proved to be the epicenter providing the best model fit, corroborating autopsy studies finding neurofibrillary tangles to start in the entorhinal cortex (Fig. 3b). Model performance was better in ADNI (global pattern $r^2 = 0.78$) compared to BioFINDER ($r^2 = 0.6$), though this difference was partially mitigated by subsampling BioFINDER to match ADNI based on demographic variables, and the difference disappeared entirely when subsampling BioFINDER to match ADNI based on mean cortical tau signal (Supplementary Fig. 3). Model fit was good across cognitively normal, MCI and AD subjects, and expected increases in mean tau signal were observed as disease severity increased (Supplementary Fig. 4). The ESM was particularly effective in predicting the early progression of tau, but diverged more from the observed tau pattern over time (Supplementary Fig. 5, Fig. 4).

As a validation, the ESM was fit over a second set of anatomical connections from another non-overlapping dataset consisting of healthy and cognitively impaired older adults. Once again, the ESM demonstrated good model fit, explaining 65.6% (null model mean $r^2$ [95% CI] = 0.107 [0.052, 0.217], $p < 0.01$) of the overall spatial pattern of tau, and explained 44.8% (SD = 21.7%; null model mean $r^2$ [95% CI] = 0.104 [0.077, 0.147], $p < 0.01$) of the spatial pattern within-individual subjects on average (Supplementary Fig. 6).

The ESM was fit once again using connectivity matrices composed of functional connections measured in separate samples

of young healthy adults, and old healthy and impaired adults, respectively, using resting-state functional MRI connectivity (Supplementary Fig. 6, Fig. 4). These analyses test whether the ESM is robust to different measures of macroscale connectivity, but also can be thought to test an alternative hypothesis of tau spread through communication of pathological states, rather than through physical spread of tau oligomers. Models fit over functional connectomes performed quite well, though slightly worse than models using structural connectomes (YOUNG: Global $r^2 = 0.565$; null $r^2$ [95% CI] = 0.089 [0.031–0.187]; individual mean $r^2 = 0.384$, SD = 0.168, null $r^2$ [95% CI] = 0.103 [0.069–0.156]; OLD: Global $r^2 = 0.586$; null $r^2$ [95% CI] = 0.031 [0.000–0.087]; individual mean $r^2 = 0.451$, SD = 0.209, null $r^2$ [95% CI] = 0.063 [0.037–0.109]), a trend that was consistent across preprocessing strategies (Supplementary Fig. 2D). Additional alternative hypotheses have been proposed suggesting tau may simply spread extracellularly across neighboring regions, rather than through anatomical connections. To test this hypothesis, a model was fit over a Euclidean distance matrix instead of a functional or structural connectome (Supplementary Fig. 6). As with models using functional connectomes, the euclidan distance matrix performed far greater than chance, but not as well as models using anatomical connectivity.

**Low-level tau spreading is evident in Aβ− individuals.** We divided our study sample into groups based on Aβ status and examined model accuracy separately within these groups. Model accuracy remained high even among Aβ− individuals, despite a low overall tau burden (Fig. 3a). These effects were additionally

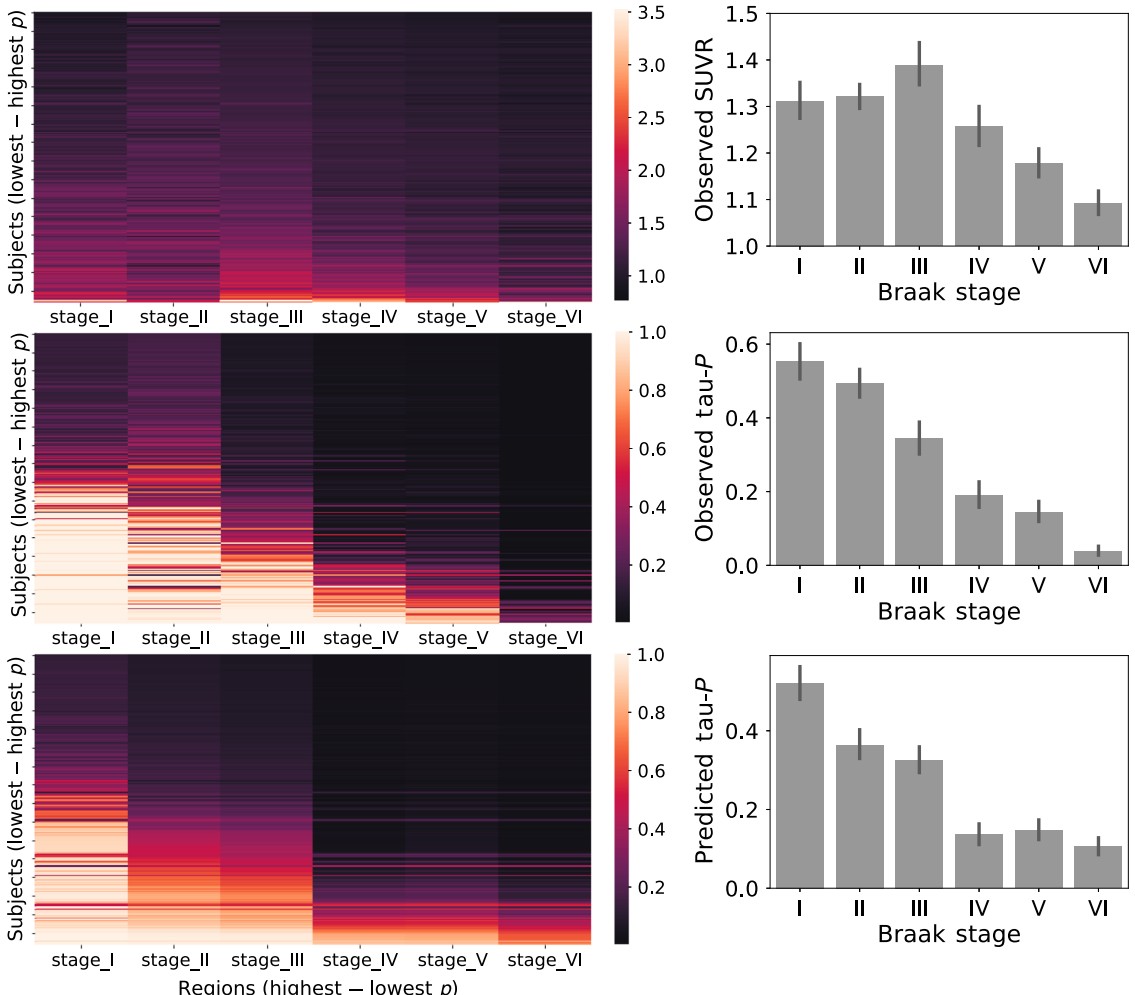

**Fig. 2 Tau-positive probabilities recapitulate Braak staging.** Each brain region was divided into one of six "Braak stage" ROIs, based on which Braak stage the region first shows abnormal tau (as described in ref. [83]). (Left) Each row is a subject sorted top-bottom by least to most overall tau. Each column is an Braak stage ROI, sorted left to right by most to least overall tau. Warmer colors represent higher SUVR values (top), observed tau-positive probabilities (middle) or predicted tau-positive probabilites from the best-fitting ESM (bottom). (Right) The same relationship shown in a barchart format. Error bars represent standard error of the mean. Conversion to tau-positive probabilities creates a sparse distribution of values demonstrating a progression reminiscent of the staging described in the autopsy literature.

present when including Aβ− MCI subjects, when summarizing within MCI− subjects alone, and when summarizing results over only cognitively normal Aβ− individuals without marginally elevated CSF Aβ and without any *APOE*4 allele copies (Fig. 5b). This was validated by examining model fit against the tau pattern of individual Aβ− subjects (Fig. 5). Model performance was high across most CN− subjects (Fig. 5a), including those with low or even very low regional tau burden (Fig. 5c). In many cases, tau levels that would otherwise be considered subthreshold none-theless demonstrated a systematic pattern resembling Braak sta-ging, which was also predicted by brain connectivity.

**Regional β-amyloid affects regional model performance.** For each model, regions of interest were was classified as either overestimated or underestimated by the model based on the sign of the residual (Fig. 6a, b). Underestimated regions are those demonstrating greater tau burden than would be expected given connectivity to the model epicenter (i.e. observed > predicted), while overestimated regions demonstrate less tau than would be expected given their connectivity profile (i.e. predicted > observed). We compared regional model performance to regional Aβ accumulation as measured from a large dataset of

Aβ− PET ($^{18}$F-florbetapir, or AV45) scans (Fig. 6c). Compared to overestimated regions, underestimated regions had greater global β-amyloid burden ($t = 2.9$, $p = 0.004$; Fig. 6d), suggesting the regional presence of Aβ may accelerate the spread or expression of tau tangles. Indeed, we observed a significant correlation ($p < 0.001$) between regional model residuals and regional Aβ levels (Fig. 6e), and this relationship remained significant when adjusting for regional tau.

**Evidence for individual asymmetry in tau deposition.** Asym-metric lateralization of tau pathology and tau-PET signal is a prominent feature of rare AD variants[40], and pathology studies have highlighted examples of hemispheric asymmetry in tau spreading[10]. We used the ESM to investigate whether a general lateralization of tau deposition could be observed across the population, or whether asymmetric patterns in tau deposition were observable at the individual level. We did not observe a trend of better model performance when using either the left or right entorhinal cortex as the sole epicenter, suggesting tau does not start preferentially in one hemisphere or the other across a population (Fig. 7a). This effect was only observable when using models fit over DTI connectomes, since rsfMRI connectomes

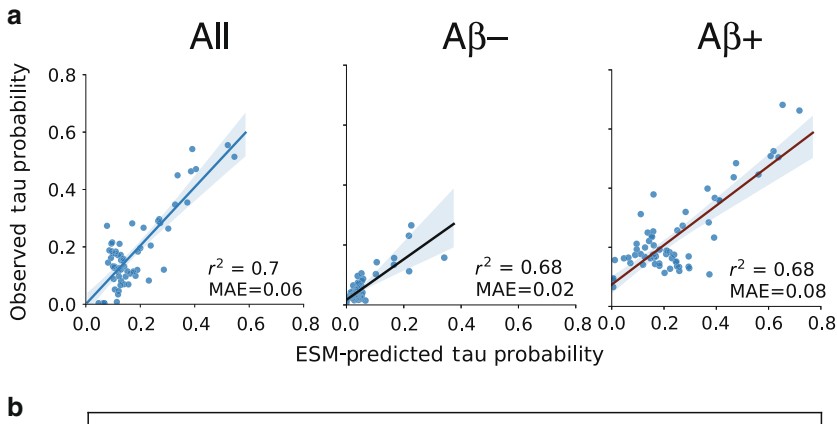

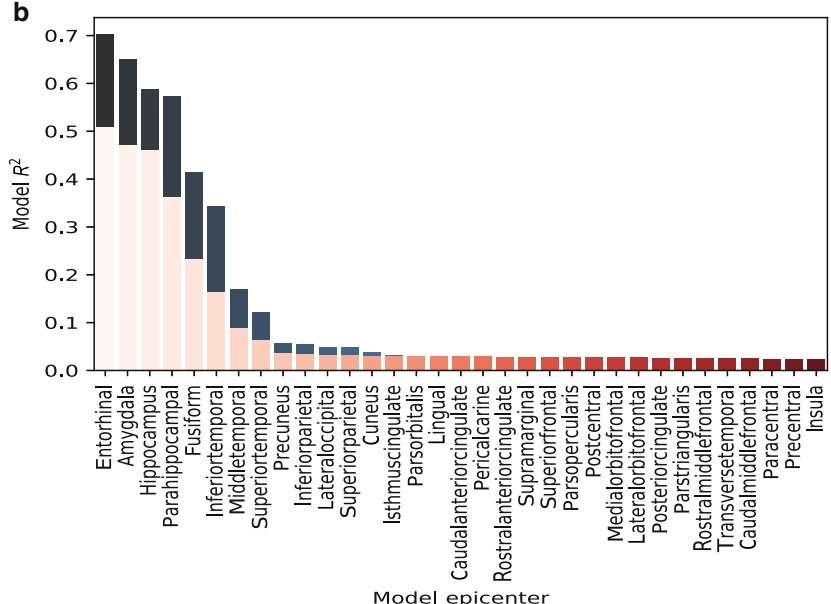

**Fig. 3 Performance of ESM in predicting spatial progression of tau. a** For each plot, each dot represents a region. The *x*-axis represents the mean simulated tau-positive probabilities across the population, while the *y*-axis represents the mean observed tau-positive probability. A value of (say) 0.3 for a given ROI would suggest that an average of 30% of all subjects included were predicted (*X*) or observed (*Y*) to have positive abnormal tau signal in that region. The average performance of the four different models are shown separately for (left) all subjects, (center) A$\beta$− individuals and (right) A$\beta$+ individuals. **b** The ESM was rerun using each left–right pair of ROIs as the model epicenter. The model fit ($r^2$) is depicted on the *y*-axis, and each bar represents the fit of a model using a given region as model epicenter. Blue bars represents global model fit across all subjects, and red bars represent the mean within-subject model fit. An entorhinal cortex epicenter provided the best model fit.

exhibited strong heterotopic (and likely indirect) connectivity in the entorhinal cortex. We next determined the best-fitting epicenter for each individual subject in the study, and categorized subjects accordingly as best described by a left-limbic, right-limbic, or non-limbic epicenter. Epicenter hemisphere was associated with asymmetry in tau deposition ($p < 0.001$), and this effect became more prominent ($ps < 0.01$) as disease severity progressed (Fig. 7d). Specifically, individuals with a left-limbic epicenter exhibited greater left temporoparietal binding, but less right frontal binding, after adjusting for disease status, age, and sex. This may point to a differing cortical expression of tau depending on the hemisphere of origin. Right-limbic epicenters were more common, but decreased with disease progression (Fig. 7b, c). Individuals with a right-limbic epicenter tended to be older ($p = 0.01$; Fig. 7d), but did not differ in sex, education, amyloid status, *APOE4* status, or total tau.

## Discussion

Observations in post-mortem human brains[26,27] and experiments in animal models[14,22–25] have together provided evidence

that tau can be transmitted from cell to cell through neuronal projections. However, post-mortem studies cannot provide direct evidence of cell-to-cell spread, and while animal models have proven tau can spread through neuronal connections under certain unnatural conditions, they cannot prove that this phenomenon occurs naturally in humans. Studies searching for evidence of cell-to-cell transmission of tau in living humans have been limited by small datasets, simplistic models, and issues relating to the quantitative measurement of tau. Here, we used a mixture-modeling approach on a large sample of humans on the Alzheimer's disease spectrum to enhance the quantification of tau signal, and we applied to this data a diffusion model based on theoretical principles of an agent propagating through a network. These simulations explained a majority of the variance in the global spatial distribution of tau-PET signal in the brain, and performed nearly as well in predicting the distribution of tau-PET signal in individual subjects. A similar model testing the hypothesis that tau spreads across neighboring brain regions was less successful at explaining the overall pattern. The models performed well in both A$\beta$-negative and A$\beta$-positive individuals, and also systematically underestimated the

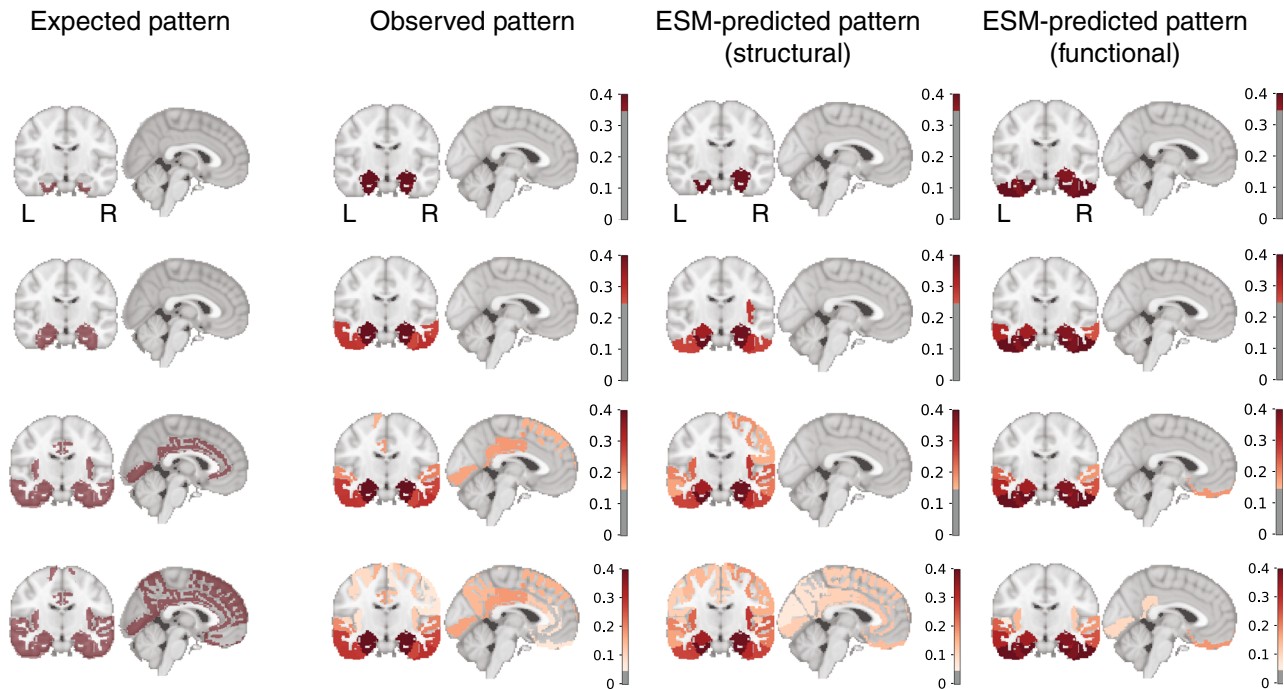

**Fig. 4 Hypothesized, observed, and predicted pattern of tau spreading.** (Left) Hypothetical spread patterns represented by Braak stages I, II, VI, V, and VI as described in ref. [83]. (Right) Spreading patterns of (from left to right) the observed tau-PET data, the ESM simulated data using a young structural connectome, and using a young functional connectome. Warmer colors represent higher proportion of regional tau-positivity predicted or observed across the population. Each "stage" was achieved by arbitrarily thresholding the population-mean tau-positive probability image at the following thresholds: 0.35, 0.25, 0.15, and 0.05.

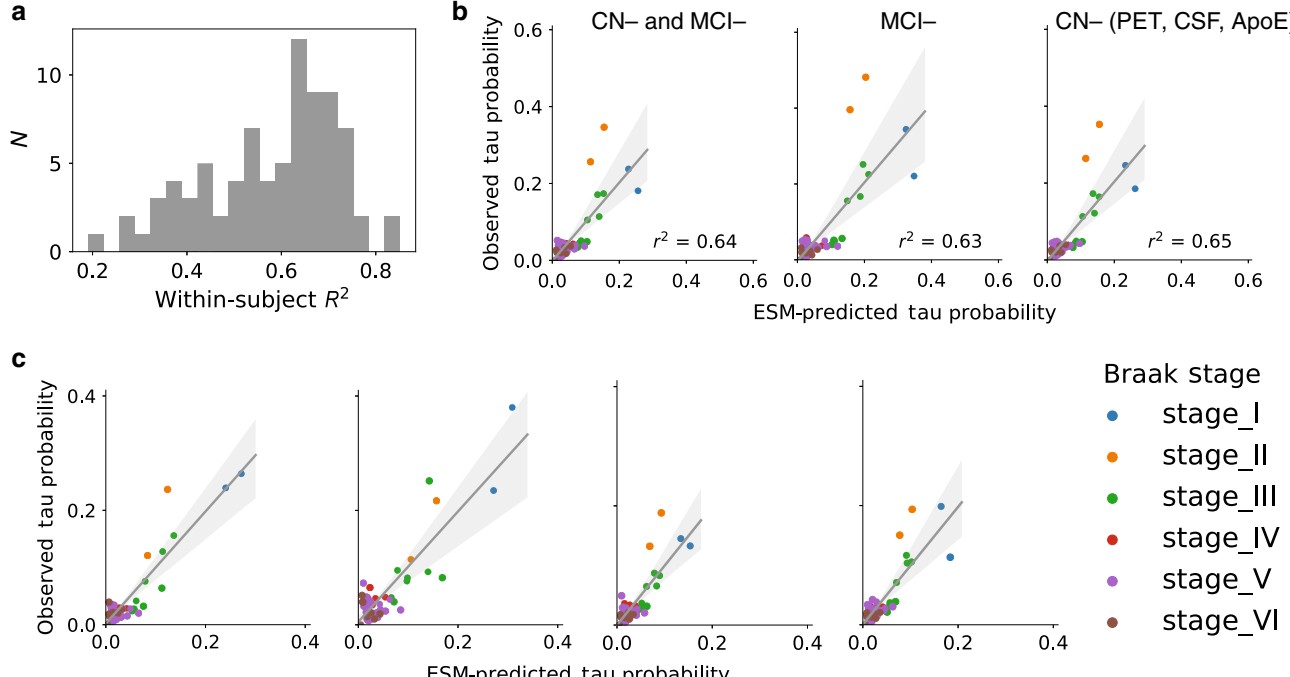

**Fig. 5 Model performance in CN− individuals.** All plots are based on the best-fitting ESM model described in the text. **a** The distribution of $r^2$ values representing the range in individual-level model fit across all CN− subjects. **b** For each plot, each dot represents a region. The x-axis represents the mean simulated tau-positive probabilities across the population, while the y-axis represents the mean observed tau-positive probability. Predicted and observed patterns are plotted for (left) all A$\beta$− individuals ($n = 104$), (middle) only A$\beta$− MCI subjects ($n = 22$), and (right) individuals without elevated A$\beta$− PET or A$\beta$− CSF, and who carry no APOE4 alleles ($n = 62$). **c** Four exemplary subjects spanning both cohorts are plotted. All four subjects are cognitively normal with MMSE 29–30 and do not carry an APOE4 allele. Their respective ages are 73, 63, 71 and 78. Even at very low (subthreshold) levels, the distribution of tau follows a pattern similar to Braak staging, and which is predicted by anatomical connectivity patterns.

 

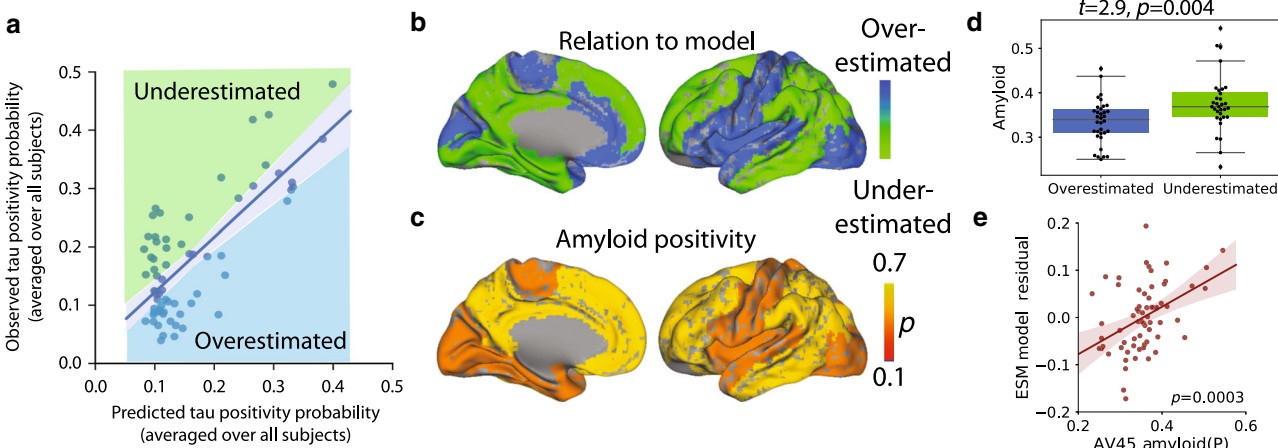

**Fig. 6 Amyloid explains regional model underestimation. a** Regions were classified as overestimated or underestimated based on the sign of the residual in a comparison of predicted vs. observed values. **b** A cortical surface render showing the spatial distribution of over- and underestimated regions. **c** A surface render showing the spatial distribution of regional amyloid-positive probabilities averaged over all subjects. **d** Underestimated regions tended to have significantly greater amyloid burden, suggesting these regions had more tau than would be predicted given their connectivity to the model epicenter. For boxplots, the center line=median, box=inner quartiles, whiskers=extent of data-distribution except *=outliers. **e** Correlation between regional model residuals and regional amyloid. Each point is a brain region, and the y-axis summarizes the degree to which a region was underestimated (positive) or overestimated (negative) by the model.

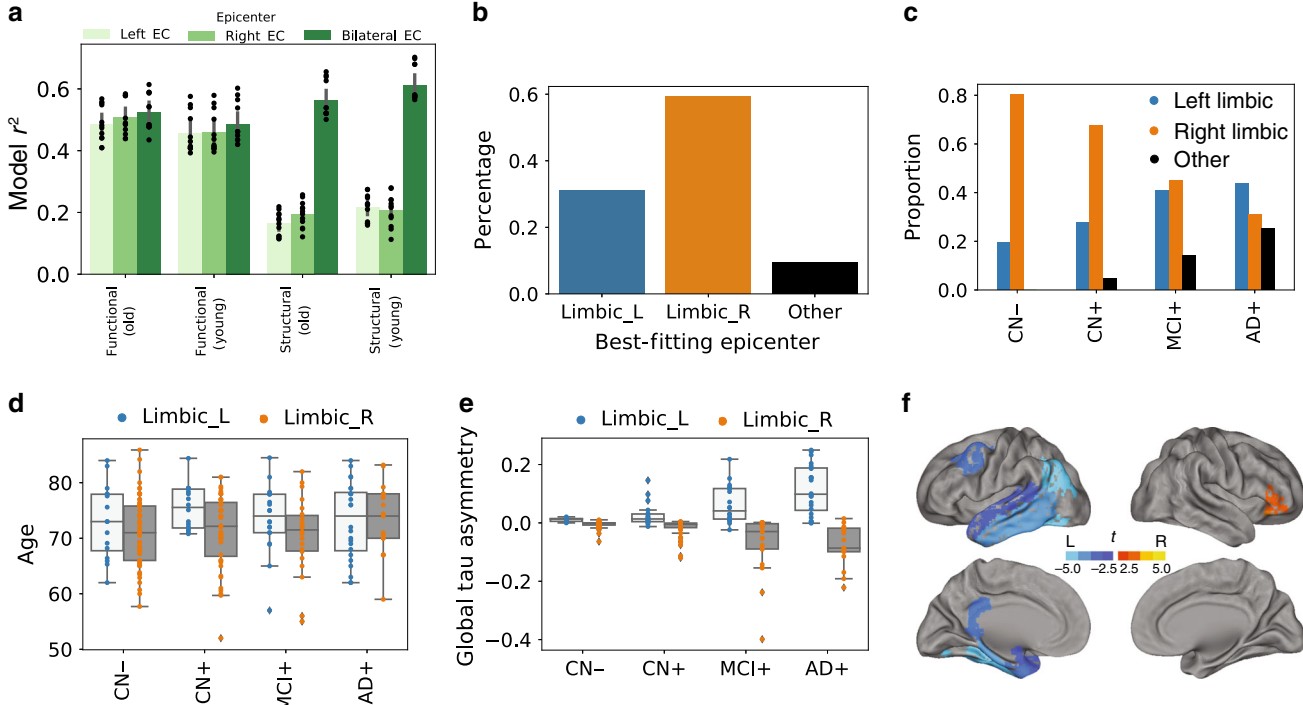

**Fig. 7 Epicenter hemisphere associated with individual variation in demographics and tau-PET binding patterns. a** Using only left or right entorhinal cortex alone as model epicenter did not result in improvement in model fit. Error bars represent standard error of the mean in variation in model fit depending on PVC strategy, confound-regression strategy, and MCI− inclusion/exclusion. **b** Proportion of individuals for whom a left-limbic, right-limbic, or cortical epicenter best fit their individual tau-PET pattern. **c** The same, across disease progression categories. **d** Subjects for whom left-limbic epicenter best fit their data were older, using a two-tailed GLM adjusting for disease status. **e** Epicenter hemisphere was associated with increasing hemispheric asymmetry in tau-PET signal across disease progression, using a two-tailed GLM adjusting for disease status. **f** Regions of higher average tau-PET signal in subjects for whom left-limbic (blue) or right-limbic (orange) epicenters best fit their data; adjusted for age, sex, disease status, and multiple comparisons. For boxplots in panels **d** and **e**: the center line=median, box=inner quartiles, whiskers=extent of data-distribution except *=outliers.

magnitude of tau in regions classically shown to harbor β-amyloid. Together, these results provides evidence that tau spreads through the limbic network in normal aging, and that the presence of β-amyloid is associated with acceleration of tau tangle expression into isocortical regions.

Brain networks may be key to the evolution of neurodegenerative disease[41]. The atrophy patterns of many neurodegenerative dementias have been shown to resemble resting-state functional brain networks[42–44], and network "hubs" are especially vulnerable to neurodegeneration across brain disorders[45]. Studies modeling

the diffusion of gray matter degeneration across brain networks have recreated such patterns with impressive accuracy[43,46,47]. However, in many neurodegenerative disorders, brain atrophy is preceded and perhaps caused by the aggregation of pathological agents. In Alzheimer's disease, the presence of tau is closely linked to[7,8], and likely precedes[48], gray matter atrophy. However, because gray matter degeneration observed in Alzheimer's dementia may be caused by many sources other than Alzheimer's pathology, gray matter degeneration itself cannot be used as proxy for tau (e.g. ref. [49]). PET studies therefore provide a unique advantage by measuring pathological proteins more directly, and applying network diffusion models to PET data has, for example, led to the successful description of the spatial progression of β-amyloid in Alzheimer's disease[50]. Our model uses a similar framework to simulate the spread of tau through the brain and reaches a similar level of success, both within-subject as well as globally across all subjects. The application of network models to other forms of dementia will be needed to conclude whether the spread of pathological proteins through connected neurons is a common thread linking many diseases.

While our model recapitulated the early stages of tau spreading accurately (Braak I–III), later stages (IV–VI) were modeled less accurately, with a systematic underestimation of tau in regions prone to early and high-volume β-amyloid aggregation. While tau, not β-amyloid, is closely associated with atrophy in Alzheimer's disease, the commonly observed concurrence of extra-limbic tau and cortical amyloid burden has led to speculation that β-amyloid may accelerate or otherwise facilitate the spread of tau outside the medial temporal lobe. Recent studies in mice have shown that β-amyloid creates an environment facilitating the rapid fibrilization of tau[14,15]. Our data support this notion, as brain regions harboring more β-amyloid, such as the precuneus and temporoparietal regions, had a higher incidence of abnormal tau than would be predicted simply by their regional connectivity to the medial temporal lobe. A conclusive model of tau spreading may not be complete without incorporating dynamic interaction with Aβ.

Tau tangles are a pathological hallmark of AD, but they are neither specific to AD nor to neurodegenerative disease in general. The process of aging appears to lead inevitably to the accumulation of tau tangles in the medial temporal lobe and occasionally beyond, a phenomenon known as primary age-related tauopathy (PART)[9]. In vivo evidence for the longitudinal accumulation of tangles in healthy elderly has been observed[11]. While PART may result in subtle insults to cognition and brain health[12,13,51], there is still debate as to whether PART and AD are distinct processes[52]. We show that even in individuals without significant Aβ burden and low (subthreshold) tau-PET signal, the spatial pattern of tau resembles early Braak staging, and can be predicted by connectivity to the entorhinal cortex. This corroborates a recent study finding tau-PET patterns overlap greater than chance with entorhinal cortex connectivity even in Aβ-negative subjects[53]. The inability of Aβ-PET to identify sparse Aβ burden, especially in cases with predominant diffuse plaques, may lead to the possibility that undetectable levels of Aβ pathology may be driving the observed relationships. However, we demonstrated an early Braak-like pattern of tau in individuals at very low likelihood of having Aβ pathology (cognitively normal, APOE4-negative, CSF Aβ negative). These findings suggest that, even in normal aging, tau may spread through communicating neurons. The results also suggest closer scrutiny of subthreshold tau-PET signal in cognitively unimpaired, Aβ-negative individuals. Elevated SUVR values occurring in a consistent pattern in specific limbic regions may be indicative of very low tau pathology, rather than non-specific or off-target ligand binding.

Tau can be directly secreted into extracellular space, and mechanisms have been described for subsequent cellular uptake (c.f. ref. [54]), leading to the hypothesis that tau may be propagated to neighboring neurons. This idea is not supported by our data, where neuronal connectivity patterns provided a better description of the in vivo spatial distribution of tau. Another hypothesis stems from the observation that tau has an excitatory effect on neurons[55], but is also secreted by activated neurons[55,56]. These two observations have lead to the idea of an excitotoxic cascade, where the presence of tau excites neurons, leading to over-stimulation of connected neurons, which in turn leads to secretion of tau, and so forth. This latter hypothesis cannot be ruled out based on our data, as it is still predicated on the spreading of pathological events across communicating neurons. In our study, we fit the ESM over two different measures of macroscale connectivity, and the choice of modality comes with different sets of assumptions and limitations. DTI tractography endeavors to directly measure white matter connections between brain regions, and may therefore be the most appropriate choice, but also suffers from important methodological limitations such as the gyral bias[57]. On the other hand, rsfMRI connectomes are conflated by indirect connectivity[57] (e.g. Fig. 7a), which does not fit with the hypothesis of direct axonal spread. Additionally, one can imagine a scenario where a region may act as a way station for tau propagation without itself expressing pathological tau due to (say) its genomic environment. Additionally, alternative hypotheses of tau propagation involving network propagation of a pathological (e.g. excitotoxic or tau overproduction) state would not necessarily require direct connections. fMRI connectivity may be thought of a proxy of some of these hypotheses. In our data, DTI tractography-based connectomes consistently showed superior model fit compared to models fit over other connectomes (Supplementary Fig. 6, Fig. 2d), once again lending support to the cell-to-cell transmission hypothesis, though model fit was ultimately high and reproducible across both connectivity modalities. Next-generation tractography may provide improved models in the future[58], but both measures of connectivity appear to be sufficient for fairly high-performing simulations of tau spread.

While our findings lend support to the hypothesis of tau spreading through communicating neurons, connectivity patterns and regional Aβ burden together could not fully explain the observed pattern of tau-PET across the brain. While a portion of this discrepancy may be explained by measurement error, there are likely other factors at play. Recent work has outlined a consistent genomic profile across regions that express tau[59], implicating that regional variation in intrinsic molecular environment may mediate the presence and rate of tau tangle formation. This may explain why, for example, many subcortical regions do not show substantial tau burden despite connections to regions expressing neurofibrillary tau tangles. In addition, it is also possible that only certain neuron types can facilitate the transmission of tau, which may be challenging to model using macroscopic neuroimaging-based measures of brain connectivity (though recent advances in single-cell transcriptomic changes in AD may help guide such analyses[60]). Heterogeneity in tau patterns[61,62] present yet another difficulty in tau spread modeling. Finally, some studies have suggested the directional flow of neuronal activity may influence the spread of brain pathology[63]. Future studies incorporating this information, along with dynamics related to regional amyloid burden and regional vulnerability, may achieve a more complete model of tau spreading. However, at present, we show that the spread of tau is predicted by connectivity patterns to a degree that greatly exceeds both chance and other hypotheses of tau spread, and does so in a parsimonious fashion, supporting the notion that connectivity is in some way involved in the spread of tau through the human brain.

The results of the ESM represent an advance on previous human studies testing the spreading hypothesis of tau. Many previous studies addressing this hypothesis have elected to examine covariance between tau patterns and brain networks, usually measured with rsfMRI. Jones et al.[33], Adams et al.[53], and Hoenig et al.[35] described overlap between data-driven tau-PET covariance networks and resting-state functional networks. Franzemeier et al.[38] and Ossenkoppele et al.[39] each went further to show correlations between rsfMRI connectivity and cross-subject covariance in tau-PET signal, within networks or across the whole brain. Sepulcre et al.[64] instead used longitudinal tau covariance across spatially distributed regions to infer connectivity between those regions. Each of these studies represent clues that tau spreading and connectivity are related in humans. However, they do not construct, test, or simulate models of tau spreading. The ESM simulates the spread of tau from the entorhinal cortex through a cascade of secondary seeding events informed by macroscale functional or structural connections, a process that is designed to mimic the hypothetical spreading of tau. This model can explain upwards of 70% of the spatial variation of tau in the human brain, representing a substantial improvement over the aforementioned associational studies, as well as over studies using similar diffusion models on structural MRI measures (e.g. refs. [49,65]). Importantly, our model is unique in finding the entorhinal cortex as the best epicenter, which corroborates autopsy findings. While our simulation explains the tau-PET data to an unprecedented degree, it is imperfect and remains indirect evidence of tau spreading. However, it also provides a first step toward a tau spreading simulation model, which can be improved, perturbed, and applied in numerous contexts. In addition, the ESM has potential as a clinical tool by estimating where tau will spread based on individual regional patterns. Knowledge of the expected pattern of tau spread will be helpful in designing regional outcome measures in future treatment trials directed against tau aggregation.

We used the ESM to conduct a preliminary analysis concerning individual variation in asymmetric hemispheric distribution of tau. We observed considerable variation in laterality of tau-PET signal across individuals, particularly in later disease states, and the dominant hemipshere was predicted by the hemisphere of the best-fitting epicenter determined by the ESM. While asymmetric tau deposition is commonly described in rare AD variants[40], our findings suggest some lateralization even in typical AD, and may be associated with differential cortical patterning of tau accumulation. Subjects with right-side dominant tau patterns tended to be older, but a more thorough analysis is necessary to uncover whether differential hemispheric lateralization of tau deposition leads to distinct phenotypes of clinical expression.

Our study comes with a number of limitations. The premise of testing the hypothesis of tau spread through communicating neurons requires that both neuronal connections and tau burden are accurately measured. We attempt to partially surmount these issues by introducing a data-driven approach for overcoming off-target and non-specific binding in Flortaucipir-PET data, and by validating our findings over different connectomes across different samples and modalities. Our mixture-modeling strategy is sensitive to sample size and composition. While it is unlikely that this phenomenon strongly affected the present findings, it is an important point worth consideration for future studies utilizing this approach to transform tau-PET data. Another limitation is raised by our choice to remove regions that do not demonstrate measurable tau burden, namely subcortical regions, from the model altogether. Certain subnuclei of subcortical structures such as the thalamus do accumulate tau pathology in Alzheimer's disease[66], though we were unable to detect such pathology, perhaps due to the resolution of our measurements. While it is

possible that subcortical structures participate in neuronal transmission of pathology without expressing the pathology itself, the current implementation of our model does not support this type of dynamic. However, while incidental measurement of indirect functional connectivity is a common critique of functional MRI, here it may pose an advantage, as functional connectivity mediated by subcortical connections may still be present in functional connectomes used for this study. Finally, we tested the ESM over a number of different pre-processing decisions, and mostly describe results of best-fitting models. It is important to note that a model that best fits our data does not necessarily equate to a model that best fits biology. However, many different pre-processing combinations produced high-performing models (Supplementary Fig. 2A), so we are confident that our results are not dependent on our pre-processing decisions.

In conclusion, our data support the notion that tau pathology itself, or information leading to the the expression of pathology, is transmitted from cell to cell in humans, principally through neuronal connections, and not extracellular space. Our findings further suggest that this phenomenon proceeds fairly ubiquitously in normal aging, and that the process is accelerated in specific brain regions demonstrating $\beta$-amyloid burden. While our cross-sectional, in vivo results cannot prove that tau spreads through neuronal connections, we show that more highly connected regions have a higher tendency to be affected sooner by tau along a specific network path cascading from the medial temporal lobe. Future models may be able to improve results by incorporating region-specific vulnerability factors, directional flow, and $A\beta$ dynamics, though contributing such information in a parsimonious way presents a difficult challenge.

## Methods

**Participants**. Participants of this study represented a selection of individuals from two large multi-center studies: the Swedish BioFinder Study (BioF; http://biofinder. se/) and the Alzheimer's Disease Neuroimaging Initiative (ADNI; http://adni.loni. usc.edu). Both studies were designed to accelerate the discovery of biomarkers indicating progression of Alzheimer's disease pathology. Participants were selected based on the following inclusion criteria: participants must (i) have a Flortaucipir-PET scan, (ii) have either a $\beta$-amyloid-PET scan (for ADNI: [18F]-Florbetapir, for BioF: [18F]-Flutemetamol) or lumbar puncture measuring CSF $A\beta1$–42. In addition, participants were required to be cognitively unimpaired, have a clinical diagnosis of mild cognitive impairment, or have a clinical diagnosis of Alzheimer's dementia with biomarker evidence of $\beta$-amyloid ($A\beta$) positivity. For both cohorts separately, PET-based $A\beta1$–42 positivity was defined using mixture modeling, as previously described[5]. For BioFINDER, $\beta$-amyloid1–42 positivity was defined as an (INNOTEST) level below 650 ng/L[67]. All participants fitting the inclusion criteria with Flortaucipir scans acquired (BioFINDER) or that were available for public download (ADNI) in May 2018 were included in this study. In total, across both studies, 162 cognitively unimpaired individuals, 89 individuals with mild cognitive impairment, and 61 amyloid-positive individuals with suspected Alzheimer's dementia were included. Demographic information can be found in Table 1, whereas a detailed comparison of BioFINDER and ADNI cohorts can be found in Supplementary Table S1. BioFINDER subjects were on average less educated than ADNI subjects, and included a higher proportion of amyloid-positive individuals. All BioFINDER subjects provided written informed consent to participate in the study according to the Declaration of Helsinki; ethical approval was given by the Ethics Committee of Lund University, Lund, Sweden, and all methods were carried out in accordance with the approved guidelines. Approval for PET imaging was obtained from the Swedish Medicines and Products Agency and the local Radiation Safety Committee at Skåne University Hospital, Sweden. Information related to participant consent in ADNI can be found at (ADNI; http://adni.loni.usc.edu).

**PET acquisition and pre-processing**. MRI and PET acquisition procedures for ADNI (http://adni.loni.usc.edu/methods/) and BioFINDER[68] have both been previously described at length. All Flortaucipir-PET scans across studies were processed using the same pipeline, which has also been previously described[34,68]. Briefly, 5-min frames were reconstructed from 80 to 100 min post-injection. These frames were re-aligned using AFNI's 3dvolreg (https://afni.nimh.nih.gov/) and averaged, and the mean image was coregistered to each subject's native space T1 image. The coregistered image was intensity normalized using an inferior cerebellar gray reference region, creating standard uptake value ratios (SUVR). In order to get an independent map of $\beta$-amyloid ($A\beta$) deposition, regional $A\beta$-PET images were downloaded from a larger cohort of subjects. Baseline ROI-level information for

[18]F-Florbetapir scans were downloaded from available ADNI subjects ($n = 974$), which had been processed using the whole cerebellum as a reference region.

**The Epidemic Spreading Model.** The spread of tau through connected brain regions was simulated using the ESM, a previously described diffusion model that has been applied to explain the spread of $\beta$-amyloid through the brain[50]. The ESM simulates the diffusion of a signal from an epicenter through a set of connected regions over time (Fig. 1a, b). The dynamics of the spreading pattern are controlled by the weighted connectivity between regions, and by a set of parameters fit within-subject, the latter of which are solved through simulation. Specifically, the parameters represent subject-specific (i) global tau production rate, (ii) global tau clearance rate, and (iii) age of onset, which interact with regional-connectivity patterns to determine the velocity of spread. The ESM is simulated over time for each subject across several parameter sets, and the set that produces the closest approximation to observed tau burden for a given subject is selected. Note that these parameters themselves do not control regional patterning, which is the metric by which the accuracy of the model is evaluated (see below). Instead, the free parameters moderate the overall tau burden (i.e. the stopping point), which allows the ESM to be fit to individuals across the Alzheimer's disease spectrum. For example, an individual with little-to-no tau burden would likely be fit with a balance of production and clearance rates that would preclude the overproduction and spread of tau signal (Fig. 1c). A detailed and formalized description of the ESM can be found elsewhere[50].

The ESM takes as input a Region × Subject matrix of values ranging from 0 to 1, representing the probability of a pathological burden (in this case, of tau) in a given region for a given subject. The model is fit within-subject and, for each subject, produces an estimate of tau probability for every region of interest. In previous applications of the ESM, the model is fit over every possible epicenter as well as combinations of epicenters, and the epicenter providing the best overall fit to the data is selected. In our case, autopsy work provides strong evidence for a consistent "epicenter" of tau neurofibrillary tangles in humans. Tangles first emerge in the transentorhinal cortex, before emerging in other parts of the entorhinal cortex as well as the anterior hippocampus[10,17]. We therefore ran models with the left and right entorhinal cortex selected as the model epicenters. In order to validate this choice, we ran the model using the left–right pair of every region of interest (33 pairs in all) and compared the model fit using each regional epicenter. To examine asymmetric spreading, we later fit models using just the left and right entorhinal cortex as separate epicenters. We also found a best-fitting model-derived epicenter for each subject, by fitting the ESM across all possible regions and finding the best within-subject fit.

There are many data pre-processing and model fitting decisions that may affect the performance of the ESM. Some of these decisions include (i) what kind of connectivity data to fit the model over, (ii) which brain regions to include, (iii) what kind of tau measurement to use as input, (iv) whether regional tau-PET data should be partial volume corrected, (v) whether and how to correct the regional tau-PET data for confounding signals, and (vi) whether or not to include amyloid-negative MCI subjects. Rather than arbitrarily choosing these parameters, we fit the ESM over a range of different parameter sets (see subsequent sections) and investigate how these pre-processing decisions affect model performance. We then select the best-fitting models for subsequent analysis. Choices for (ii)–(v) are discussed in Section "Regional tau-PET data pre-processing", whereas choices for (i) are discussed in Section "Connectivity measurements". Across all combinations of methodological choices, a total of 432 models were fit.

**Regional tau-PET data pre-processing.** Preprocessing of PET data resulted in mean regional tau-PET SUVR values from the FreeSurfer-derived Desikan-Killiany-Tourville (DKT) atlas[69], extracted from each individual's native space PET image. Only cortical and subcortical regions overlapping with the MindBoggle DKT atlas were used[70], leaving 78 regions in total. Previous Flortaucipir-PET studies have noted considerable off-target binding of the Flortaucipir signal, leading to signal in regions without pathological tau burden, and likely to pollution of signal in regions accumulating tau[28,29,31,32,34]. While many previous studies have ignored these issues, accounting for off-target binding is essential to the current study, as our model cannot distinguish off-target from target signal, and we are not interested in the propagation of off-target signal. To address this issue, we utilized regional Gaussian mixture modeling under the assumption that the target and off-target signal across the population are distinct and separable Gaussian distributions (Fig. 1c).

As most individuals do not have tau in most regions, pathological signal should show a skewed distribution across the population, whereas off-target and non-specific signal should be reasonably normally distributed. Such a bimodal distribution has been observed for $\beta$-amyloid, and mixture modeling has been used in this context to define global $\beta$-amyloid positivity[71,72]. Our approach differs from these previous studies as we do not assume the distribution of target and off-target binding to be homogeneous across cortical areas—we apply Gaussian mixture modeling separately to each region of interest (Fig. 1c). Specifically, for each region, we fit a one-component and a two-component Gaussian mixture model across the entire population. We compare the fit of the two models using Aikake's information criterion. If a two-component model fits the data better, this likely indicates the presence of pathological tau in a proportion of the population, and the

Gaussians fit to the data provide a rough estimate of an SUVR threshold, above which Flortaucipir signal has a high probability of being abnormal. If a one-component model fits better, this indicates the Flortaucipir-PET signal within the region is roughly normally distributed across the population, which we do not expect for tau in a population including many cognitively impaired individuals. The ESM receives regional (tau) probabilities as input, and so we calculate the probability that a given subject's ROI SUVR value falls onto the second (i.e. right-most) Gaussian distribution using repeated fivefold cross-validation. Assuming this second distribution represents the subjects with abnormal Flortaucipir signal, this value estimates the proximity of a subject to the pathological distribution. Effectively, this converts regional SUVRs to regional tau-positive probabilities. This approach defines a fairly conservative, data-driven threshold for SUVR values, above which, one can assume the presence of abnormal signal (perhaps indicating pathological tau accumulation) with a high degree of confidence.

For purposes of comparison, we also use two other preprocessing strategies for regional tau-PET data. First, we apply a regional normalization of SUVR values along a 0–1 scale, which is equivalent to simply using SUVR values as input (the ESM expects values to be between 0 and 1). Second, we reproduce the reference strategy described in the original ESM paper. This approach involves creating a null distribution by obtaining the maximum value of 40,000 bootstrapped samples of the 5–95% largest SUVR values within the reference region. The distribution is used to create an empirical cumulative distribution function, which is applied to each voxel of the PET image, effectively finding the probability that this voxel is greater than values in the reference region (see ref. [50] for details). We also fit the model using different region-sets: (i) all cortical and subcortical regions ($n = 78$), (ii) cortical regions only (including hippocampus and amygdala, $n = 66$), (iii) only regions demonstrating a bimodal distribution ($n$ varies depending on other pre-processing decisions).

As mentioned above, tau-PET signal is confounded by a number of off-target binding sources, some of which are age related[28,32]. Some studies have found that regressing out certain signal sources, such as choroid plexus binding or age-related subcortical signal, can improve expected relationships between Flortaucipir and other measures (e.g. ref. [73]). In addition, recent studies have found a putative impact of sex on Flortaucipir binding[74,75]. Therefore, we explored the impact of removing confounding signals from tau-PET data on model performance. We tried three different strategies: (i) no preprocessing, (ii) regressing out age, sex and mean choroid plexus binding from each region separately across all subjects, (iii) using a $W$-score approach[76], where regional SUVR values are normalized by A$\beta$-negative cognitively normal elderly adjusting for age, sex, and choroid plexus binding. Native space choroid plexus regions were available for each subject from the Freesurfer parcellation, and the mean Flortaucipir signal was taken between left and right hemispheres. In addition to these processing steps, we experimented with the choice of partial volume correcting (PVC) data before running the model. The geometric transfer matrix[77] method was used for PVC, and models were run with and without PVC.

**Connectivity measurements.** The overall pattern of spread simulated by the ESM is determined by the relationship matrix, which represents pairwise relationships between each region of interest. Indeed, this is the system through which the simulated signal will diffuse. Varying the relationship matrix can, for example, allow for tests of different hypotheses of spread. In addition, replicating model effects over different connectomes can improve confidence that results are robust to different samples or modalities. We fit the ESM over four different connectivity datasets, none of which overlap with one another or with subjects from the tau-PET dataset. We use anatomical connectivity measurements generated using DTI tractography from (i) healthy and impaired older adults and (ii) young healthy adults. We further validate this procedure using a functional connectivity matrix generated from (iii) healthy and impaired older adults and (iv) young healthy controls to test the hypothesis that tau spreads through communicating neurons. Finally, we additionally test the hypothesis of tau spreading through extracellular space by using a Euclidian distance matrix as input.

We created two template structural connectivity matrices using DTI tractography data from two different samples. The first was a dataset of 60 young healthy subjects from the CMU-60 DSI Template[78] (http://www.psy.cmu.edu/coaxlab/data.html). The second was a sample of healthy older and cognitively impaired older adults from ADNI. Demographic information and comparisons to other datasets can be found in Supplementary Table S1. In total, 204 individuals had one or more DTI scans available, for a total of 540 scans. The two datasets were preprocessed separately with a previously described diffusion tractography pipeline[79], and acquisition and processing information has been described in detail[80]. Briefly, orientation distribution functions (ODF) were calculated and in turn used to generate deterministic connections between pairs of brain regions from the Desikan atlas. Specifically, an ACD measure was used, representing the total proportion of regional surface area (across both regions) that contain connecting fibers between the two regions. All images were assessed for quality. Connectomes were averaged across all subjects within each template, resulting in a template structural connectome in aging and in health, respectively.

Functional connectivity measurements were generated separately from two different datasets. The first was a subsample of young healthy controls from the COBRE dataset[81], a publicly available sample which we accessed through the

Nilearn python library. All subjects listed as healthy controls under the age of 40 were selected, totaling 74 individuals. The images were already preprocessed using the NIAK resting-state pipeline (http://niak.simexp-lab.org/pipe_preprocessing.html), and additional details can be found elsewhere[81]. The second dataset consisted of a subsample of 189 healthy and cognitively impaired older adults from ADNI who passed quality control procedures. Demographic data and comparison to the other datasets can be found in Supplementary Table S1. These data were processed in-house using NIAK in a manner described previously[82]. Separately for each dataset, correlation matrices were generated by finding the correlation between timeseries' of each pair of regions of interest from the Desikan-Killiany atlas, and all available confounds were regressed from the correlation matrices. We took the mean of all correlation matrices to create an average healthy connectome template, and an average older/impaired connectome template. These connectomes were then thresholded so as to only retain the top 10% of connections, and transformed so all values fell between 0 and 1.

To create a Euclidian distance matrix, we calculated the coordinate representing the center of mass for each region of interest, and found the Euclidian distance between it and the center of mass of every other ROI. The matrix was normalized to a 0-1 scale and inverted. By using this distance matrix in the ESM, we test the hypothesis that tau diffuses radially across adjacent cortex, rather than through connected regions.

**Statistical analysis**. The ESM was fit using different relationship matrices and across several different preprocessing choices (see above). Each model was evaluated by mean within-individual fit, as well as global population fit. Individual model fit is calculated as the $r^2$ between predicted regional tau probabilities and actual regional tau probabilities measured with Flortaucipir-PET, for each individual. The mean $r^2$ across all individuals was used to represent overall model fit. To evaluate the accuracy of the global pattern, the regional predicted and observed tau probabilities, respectively, were averaged across all subjects, and the $r^2$ between these group-averaged patterns were calculated. Together, these two accuracy measures represent the degree to which regional connectivity predicts the spatial pattern of tau-PET measured within and across subjects, respectively. To ensure the magnitude of our results were greater than chance given a matrix of similar properties, for select models, we fit the ESM using 100 null matrices with preserved degree and strength distributions using the Brain Connectivity toolbox (https://sites.google.com/site/bctnet/). We use the null distribution to calculate the mean and 95% confidence intervals of the relationship occurring by chance. Since we run only 100 null models per test, the lowest possible $p$ value is 0.01, which would suggest the observed test value was higher than all values observed by chance.

To examine the global accuracy of the ESM stratified by amyloid status, we first divided all subjects into one of two diagnostic groups: amyloid negative and amyloid positive. We then calculated the mean of predicted and observed values across all subjects within each amyloid group, respectively. We performed similar analyses across different diagnoses (CN, MCI, AD). In the same manner, we also examined ESM accuracy stratified by cohort to ensure the model fit was consistent between the ADNI and BioFINDER cohorts. As a follow-up, we implemented a neighborhood search using the ball tree method and Minkowski distance ($p = 2$) to created a subsample of BioFINDER subjects matched to ADNI subjects on either demographics (Age, Sex, Education, APOE4 status) or tau load (average cortical tau-PET signal). We then once again compared model fit within this BioFINDER-matched-to-ADNI sample to model fit in ADNI subjects.

Studies in rodents have suggested a role of amyloid in facilitating the rapid fibrillarization of tau oligomers[14]. This would suggest that amyloid may play a role in explaining tau patterns that is at least partially independent of connectivity patterns. To explore this, we tested the relationship between regional modeling error and regional amyloid deposition. We converted regional amyloid SUVR values to amyloid-positive probabilities using the same regional mixture-modeling approach as described above. Next, we used the sign of the residual to divide regions into those that were overestimated by the ESM, and those that were underestimated by the ESM. An underestimated region, for example, would show more tau than the model predicted given that region's connectivity to the model epicenter. We explored the relationship between model estimation and amyloid by comparing the degree of (group-mean) amyloid between overestimated and underestimated regions using $t$-tests. We also calculate the correlation between regional model residuals and regional amyloid values. To ensure this relationship is independent of local tau, we fit a model assessing the independent relationship of regional amyloid and tau, respectively, on regional model residuals.

To investigate global asymmetry in tau spreading, we compared the performance of ESM fit with a left entorhinal context epicenter to performance of models fit with a right entorhinal cortex epicenter. To explore asymmetry in individual patterning, we fit the ESM over every possible epicenter and stored information pertaining to the best-fitting epicenter for each subject. Epicenters were broadly characterized into left and right hemisphere and limbic or non-limbic. Limbic epicenters included entorhinal cortex, hippocampus, amygdala, or parahippocampal gyrus. We stratified subjects by their epicenter hemisphere (Limbic-Left, Limbic-Right, Other) and used ordinary least-squares general linear models (GLMs) to examine associations between epicenter hemisphere and other covariates (age, sex, education, APOE4 status) covarying for disease status (CN−, CN+, MCI+, AD+). We also compared subjects by their total tau

asymmetry (mean of left minus right across all cortical ROIs). Finally, we ran separate GLMs assessing relationships between epicenter hemisphere and tau signal in each region of interest, covarying for disease status, age, and sex. These relationships were subsequently FDR corrected using the Benjamini–Hochberg approach.

**Reporting summary**. Further information on research design is available in the Nature Research Reporting Summary linked to this article.

## Data availability

Analyses in this manuscript were conducted principally using subjects from the ADNI and BioFINDER cohort. ADNI is a publicly available dataset and can be accessed at http://adni.loni.usc.edu/. BioFINDER data are not publicly available for download, but access requests can be made to the study Principal Investigator, Oskar Hansson. Additionally, data used to create template connectomes are also publicly available. ADNI rsfMRI and DTI data can be downloaded at http://adni.loni.usc.edu/. The COBRE dataset can be accessed at ref. [81], or can be downloaded using the Nilearn python package https://nilearn.github.io/. CMU60 DTI data can be accessed at http://www.psy.cmu.edu/coaxlab/data.html. Data used in preparation of this article were obtained from the Alzheimer's Disease Neuroimaging Initiative (ADNI) database (http://adni.loni.usc.edu).

## Code availability

Matlab scripts for the Epidemic Spreading Model will be made available in a forthcoming public software release. Inquiries into acquiring the scripts beforehand can be sent to Yasser Iturria-Medina. Python functions used in part to analyze and plot ESM data can be found at https://github.com/illdopejake/data_driven_pathology/blob/master/esm/ESM_utils.py.

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

## Acknowledgements

We would like to thank Bratislav Misic, Pierre Bellec, and Mallar Chakravarty for comments and suggestions during the formulation of this work. J.W.V. is supported by the government of Canada through the tri-council Vanier Canada Graduate Doctoral Fellowship. We would also like to acknowledge support from the Ludmer Centre for Neuroinformatics and Mental Health and the Healthy Brains for Healthy Lives initiative. Work at the authors' research center was supported by the European Research Council, the Swedish Research Council, the Knut and Alice Wallenberg foundation, the Marianne and Marcus Wallenberg foundation, the Strategic Research Area MultiPark (Multidisciplinary Research in Parkinson's disease) at Lund University, the Swedish Alzheimer Foundation, the Swedish Brain Foundation, The Parkinson foundation of Sweden, The Parkinson Research Foundation, the Skåne University Hospital Foundation, and the Swedish federal government under the ALF agreement. Doses of 18F-flutemetamol injection were sponsored by GE Healthcare. The precursor of 18F-flortaucipir was provided by AVID radiopharmaceuticals. Data collection and sharing for this project was funded by the Alzheimer's Disease Neuroimaging Initiative (ADNI) (National Institutes of Health Grant U01 AG024904) and DOD ADNI (Department of Defense award number W81XWH-12-2-0012). ADNI is funded by the National Institute on Aging, the National Institute of Biomedical Imaging and Bioengineering, and through generous contributions from the following: AbbVie, Alzheimer's Association; Alzheimer's Drug Discovery Foundation; Araclon Biotech; BioClinica, Inc.; Biogen; Bristol-Myers Squibb Company; CereSpir, Inc.; Cogstate; Eisai Inc.; Elan Pharmaceuticals, Inc.; Eli Lilly and Company; EuroImmun; F. Hoffmann-La Roche Ltd and its affiliated company Genentech, Inc.; Fujirebio; GE Healthcare; IXICO Ltd; Janssen Alzheimer Immunotherapy Research & Development, LLC.; Johnson & Johnson Pharmaceutical Research & Development LLC.; Lumosity; Lundbeck; Merck & Co., Inc.; Meso Scale Diagnostics, LLC.; NeuroRx Research; Neurotrack Technologies; Novartis Pharmaceuticals Corporation; Pfizer Inc.; Piramal Imaging; Servier; Takeda Pharmaceutical Company; and Transition Therapeutics. The Canadian Institutes of Health Research is providing funds to support ADNI clinical sites in Canada. Private sector contributions are facilitated by the Foundation for the National Institutes of Health (www.fnih.org). The grantee organization is the Northern California Institute for Research and Education, and the study is coordinated by the Alzheimer's Therapeutic Research Institute at the University of Southern California. ADNI data are disseminated by the Laboratory for Neuro Imaging at the University of Southern California. Data used in preparation of this article were obtained from the Alzheimer's Disease Neuroimaging Initiative (ADNI) database (http://adni.loni.usc.edu). As such, the investigators within the ADNI contributed to the design and implementation of ADNI and/or provided data but did not participate in analysis or writing of this report. A complete list of ADNI investigators can be found at: http://adni.loni.usc.edu/wp-content/uploads/howtoapply/ADNIAcknowledgementList.pdf. Open access funding provided by Lund University.

## Author contributions

J.W.V., Y.I.-M., and A.C.E conceptualized the study. Y.I.-M. designed the Epidemic Spreading Model. J.W.V., Y.I.-M., and E.L. designed and developed the other methodologies. O.T.S. and R.S. preprocessed the data. J.W.V. analyzed the data. O.H. provided patient data. J.W.V. and O.H. wrote the manuscript. J.W.V., Y.I.M., R.S., A.C.E., and O.H. interpreted the findings. All authors revised the manuscript and provided critical feedback. O.H. and A.C.E. supervised the study.

## Competing interests

O.H. has acquired research support (for the institution) from Roche, GE Healthcare, Biogen, AVID Radiopharmaceuticals, Fujirebio, and Euroimmun. In the past 2 years, he has received consultancy/speaker fees (paid to the institution) from Biogen, Roche, and Fujirebio.

## Additional information

## Alzheimer's Disease Neuroimaging Initiative

Michael Weiner[4], Paul Aisen[5], Ronald Petersen[6], Clifford R. Jack Jr.[6], William Jagust[7], John Q. Trojanowki[8], Arthur W. Toga[9], Laurel Beckett[10], Robert C. Green[11], Andrew J. Saykin[12], John Morris[13], Leslie M. Shaw[14], Enchi Liu[15], Tom Montine[16], Ronald G. Thomas[5], Michael Donohue[5], Sarah Walter[5], Devon Gessert[5], Tamie Sather[5], Gus Jiminez[5], Danielle Harvey[10], Michael Donohue[5], Matthew Bernstein[6], Nick Fox[17], Paul Thompson[18], Norbert Schuff[19], Charles DeCArli[10], Bret Borowski[6], Jeff Gunter[6], Matt Senjem[6], Prashanthi Vemuri[6], David Jones[6], Kejal Kantarci[6], Chad Ward[6], Robert A. Koeppe[20], Norm Foster[21], Eric M. Reiman[22], Kewei Chen[22], Chet Mathis[23], Susan Landau[7], Nigel J. Cairns[13], Erin Householder[13], Lisa Taylor Reinwald[13], Virginia Lee[24], Magdalena Korecka[24], Michal Figurski[24], Karen Crawford[9], Scott Neu[9], Tatiana M. Foroud[12], Steven Potkin[25], Li Shen[12], Faber Kelley[12], Sungeun Kim[12], Kwangsik Nho[12], Zaven Kachaturian[26], Richard Frank[27], Peter J. Snyder[28], Susan Molchan[29], Jeffrey Kaye[30], Joseph Quinn[30],

Betty Lind[30], Raina Carter[30], Sara Dolen[30], Lon S. Schneider[31], Sonia Pawluczyk[31], Mauricio Beccera[31], Liberty Teodoro[31], Bryan M. Spann[31], James Brewer[32], Helen Vanderswag[32], Adam Fleisher[22], Judith L. Heidebrink[20], Joanne L. Lord[20], Ronald Petersen[6], Sara S. Mason[6], Colleen S. Albers[6], David Knopman[6], Kris Johnson[6], Rachelle S. Doody[33], Javier Villanueva Meyer[33], Munir Chowdhury[33], Susan Rountree[33], Mimi Dang[33], Yaakov Stern[34], Lawrence S. Honig[34], Karen L. Bell[34], Beau Ances[35], John C. Morris[35], Maria Carroll[35], Sue Leon[35], Erin Householder[13], Mark A. Mintun[35], Stacy Schneider[35], Angela OliverNG[36], Randall Griffith[36], David Clark[36], David Geldmacher[36], John Brockington[36], Erik Roberson[36], Hillel Grossman[37], Effie Mitsis[37], Leyla de Toledo-Morrell[38], Raj C. Shah[38], Ranjan Duara[39], Daniel Varon[39], Maria T. Greig[39], Peggy Roberts[39], Marilyn Albert[40], Chiadi Onyike[40], Daniel D'Agostino II[40], Stephanie Kielb[40], James E. Galvin[41], Dana M. Pogorelec[41], Brittany Cerbone[41], Christina A. Michel[41], Henry Rusinek[41], Mony J. de Leon[41], Lidia Glodzik[41], Susan De Santi[41], P. Murali Doraiswamy[42], Jeffrey R. Petrella[42], Terence Z. Wong[42], Steven E. Arnold[14], Jason H. Karlawish[14], David Wolk[14], Charles D. Smith[43], Greg Jicha[43], Peter Hardy[43], Partha Sinha[43], Elizabeth Oates[43], Gary Conrad[43], Oscar L. Lopez[23], MaryAnn Oakley[23], Donna M. Simpson[23], Anton P. Porsteinsson[44], Bonnie S. Goldstein[44], Kim Martin[44], Kelly M. Makino[44], M. Saleem Ismail[44], Connie Brand[44], Ruth A. Mulnard[45], Gaby Thai[45], Catherine Mc Adams Ortiz[45], Kyle Womack[46], Dana Mathews[46], Mary Quiceno[46], Ramon Diaz Arrastia[46], Richard King[46], Myron Weiner[46], Kristen Martin Cook[46], Michael DeVous[46], Allan I. Levey[47], James J. Lah[47], Janet S. Cellar[47], Jeffrey M. Burns[48], Heather S. Anderson[48], Russell H. Swerdlow[48], Liana Apostolova[49], Kathleen Tingus[49], Ellen Woo[49], Daniel H. S. Silverman[49], Po H. Lu[49], George Bartzokis[49], Neill R. Graff Radford[50], Francine Parfitt[50], Tracy Kendall[50], Heather Johnson[50], Martin R. Farlow[12], Ann Marie Hake[12], Brandy R. Matthews[12], Scott Herring[12], Cynthia Hunt[12], Christopher H. van Dyck[51], Richard E. Carson[51], Martha G. MacAvoy[51], Howard Chertkow[52], Howard Bergman[52], Chris Hosein[52], Sandra Black[53], Bojana Stefanovic[53], Curtis Caldwell[53], Ging Yuek Robin Hsiung[54], Howard Feldman[54], Benita Mudge[54], Michele Assaly Past[54], Andrew Kertesz[55], John Rogers[55], Dick Trost[55], Charles Bernick[56], Donna Munic[56], Diana Kerwin[57], Marek Marsel Mesulam[57], Kristine Lipowski[57], Chuang Kuo Wu[57], Nancy Johnson[57], Carl Sadowsky[58], Walter Martinez[58], Teresa Villena[58], Raymond Scott Turner[59], Kathleen Johnson[59], Brigid Reynolds[59], Reisa A. Sperling[60], Keith A. Johnson[60], Gad Marshall[60], Meghan Frey[60], Jerome Yesavage[61], Joy L. Taylor[61], Barton Lane[61], Allyson Rosen[61], Jared Tinklenberg[61], Marwan N. Sabbagh[62], Christine M. Belden[62], Sandra A. Jacobson[62], Sherye A. Sirrel[62], Neil Kowall[63], Ronald Killiany[63], Andrew E. Budson[63], Alexander Norbash[63], Patricia Lynn Johnson[63], Thomas O. Obisesan[64], Saba Wolday[64], Joanne Allard[64], Alan Lerner[65], Paula Ogrocki[65], Leon Hudson[65], Evan Fletcher[66], Owen Carmichael[66], John Olichney[66], Charles DeCarli[66], Smita Kittur[67], Michael Borrie[68], T. Y. Lee[68], Rob Bartha[68], Sterling Johnson[69], Sanjay Asthana[69], Cynthia M. Carlsson[69], Steven G. Potkin[70], Adrian Preda[70], Dana Nguyen[70], Pierre Tariot[22], Adam Fleisher[22], Stephanie Reeder[22], Vernice Bates[71], Horacio Capote[71], Michelle Rainka[71], Douglas W. Scharre[72], Maria Kataki[72], Anahita Adeli[72], Earl A. Zimmerman[73], Dzintra Celmins[73], Alice D. Brown[73], Godfrey D. Pearlson[74], Karen Blank[74], Karen Anderson[74], Robert B. Santulli[75], Tamar J. Kitzmiller[75], Eben S. Schwartz[75], Kaycee M. SinkS[76], Jeff D. Williamson[76], Pradeep Garg[76], Franklin Watkins[76], Brian R. Ott[77], Henry Querfurth[77], Geoffrey Tremont[77], Stephen Salloway[78], Paul Malloy[78], Stephen Correia[78], Howard J. Rosen[4], Bruce L. Miller[4], Jacobo Mintzer[79], Kenneth Spicer[79], David Bachman[79], Elizabether Finger[80], Stephen Pasternak[80], Irina Rachinsky[80], John Rogers[55], Andrew Kertesz[55], Dick Drost[80], Nunzio Pomara[81], Raymundo Hernando[81], Antero Sarrael[81], Susan K. Schultz[82], Laura L. Boles Ponto[82], Hyungsub Shim[82], Karen Elizabeth Smith[82], Norman Relkin[83], Gloria Chaing[83], Lisa Raudin[83], Amanda Smith[84], Kristin Fargher[84] & Balebail Ashok Raj[84]

[4]UC San Francisco, San Francisco, CA, USA. [5]UC San Diego, San Diego, CA, USA. [6]Mayo Clinic, Rochester, NY, USA. [7]UC Berkeley, Berkeley, CA, USA. [8]U Pennsylvania, Pennsylvania, CA, USA. [9]USC, Los Angeles, CA, USA. [10]UC Davis, Davis, CA, USA. [11]Brigham and Women's Hospital, Harvard Medical School, Boston, MA, USA. [12]Indiana University, Bloomington, IN, USA. [13]Washington University St. Louis, St. Louis, MO, USA. [14]University of Pennsylvania, Philadelphia, PA, USA. [15]Janssen Alzheimer Immunotherapy, South San Francisco, CA, USA. [16]University of Washington, Seattle, WA, USA. [17]University of London, London, UK. [18]USC School of Medicine, Los Angeles, CA, USA. [19]UCSF MRI, San Francisco, CA, USA. [20]University of Michigan, Ann Arbor, MI, USA. [21]University of Utah, Salt Lake City, UT, USA. [22]Banner Alzheimer's Institute, Phoenix, AZ, USA. [23]University of Pittsburgh, Pittsburgh, PA, USA. [24]UPenn School of Medicine, Philadelphia, PA, USA. [25]UC Irvine, Newport Beach, CA, USA. [26]Khachaturian, Radebaugh & Associates, Inc and Alzheimer's Association's Ronald and Nancy Reagan's Research Institute, Chicago, IL, USA. [27]General Electric, Boston, MA, USA. [28]Brown University, Providence, RI, USA. [29]National Institute on Aging/National Institutes of Health, Bethesda, MD, USA. [30]Oregon Health and Science University, Portland, OR, USA. [31]University of Southern California, Los Angeles, CA, USA. [32]University of California San Diego, San Diego, CA, USA. [33]Baylor College of Medicine, Houston, TX, USA. [34]Columbia University Medical Center, New York, NY, USA. [35]Washington University, St. Louis, MO, USA. [36]University of Alabama Birmingham, Birmingham, MO, USA. [37]Mount Sinai School of Medicine, New York, NY, USA. [38]Rush University Medical Center, Chicago, IL, USA. [39]Wien Center, Vienna, Austria. [40]Johns Hopkins University, Baltimore, MD, USA. [41]New York University, New York, NY, USA. [42]Duke University Medical Center, Durham, NC, USA. [43]University of Kentucky, city of Lexington, NC, USA. [44]University of Rochester Medical Center, Rochester, NY, USA. [45]University of California, Irvine, CA, USA. [46]University of Texas Southwestern Medical School, Dallas, TX, USA. [47]Emory University, Atlanta, GA, USA. [48]University of Kansas, Medical Center, Lawrence, KS, USA. [49]University of California, Los Angeles, CA, USA. [50]Mayo Clinic, Jacksonville, FL, USA. [51]Yale University School of Medicine, New Haven, CT, USA. [52]McGill Univ., Montreal Jewish General Hospital, Montreal, WI, USA. [53]Sunnybrook Health Sciences, Toronto, ON, Canada. [54]U.B.C. Clinic for AD & Related Disorders, British Columbia, BC, Canada. [55]Cognitive Neurology St. Joseph's, Toronto, ON, Canada. [56]Cleveland Clinic Lou Ruvo Center for Brain Health, Las Vegas, NV, USA. [57]Northwestern University, Evanston, IL, USA. [58]Premiere Research Inst Palm Beach Neurology, West Palm Beach, FL, USA. [59]Georgetown University Medical Center, Washington, DC, USA. [60]Brigham and Women's Hospital, Boston, MA, USA. [61]Stanford University, Santa Clara County, CA, USA. [62]Banner Sun Health Research Institute, Sun City, AZ, USA. [63]Boston University, Boston, MA, USA. [64]Howard University, Washington, DC, USA. [65]Case Western Reserve University, Cleveland, OH, USA. [66]University of California, Davis Sacramento, CA, USA. [67]Neurological Care of CNY, New York, NY, USA. [68]Parkwood Hospital, Parkwood, CA, USA. [69]University of Wisconsin, Madison, WI, USA. [70]University of California, Irvine BIC, Irvine, CA, USA. [71]Dent Neurologic Institute, Amherst, MA, USA. [72]Ohio State University, Columbus, OH, USA. [73]Albany Medical College, Albany, NY, USA. [74]Hartford Hosp, Olin Neuropsychiatry Research Center, Hartford, CT, USA. [75]Dartmouth Hitchcock Medical Center, Albany, NY, USA. [76]Wake Forest University Health Sciences, Winston-Salem, NC, USA. [77]Rhode Island Hospital, Rhode Island, USA. [78]Butler Hospital, Providence, RI, USA. [79]Medical University South Carolina, Charleston, SC, USA. [80]St. Joseph's Health Care, Toronto, Canada. [81]Nathan Kline Institute, Orangeburg, SC, USA. [82]University of Iowa College of Medicine, Iowa City, IA, USA. [83]Cornell University, Ithaca, NY, USA. [84]University of South Florida, USF Health Byrd Alzheimer's Institute, Tampa, FL 33613, USA.

## the Swedish BioFinder Study

Emelie Andersson[2], David Berron[2], Elin Byman[2], Tone Sundberg-Brorsson[2], Administrator[2], Emma Borland[2], Anna Callmer[2], Cecilia Dahl[2], Eske Gertje[2], Anna-Märta Gustavsson[2], Joanna Grzegorska[2], Sara Hall[2], Oskar Hansson[2], Philip Insel[2], Shorena Janelidze[2], Maurits Johansson[2], Helena Sletten[2], Jonas Jester-Broms[2], Elisabet Londos[2], Niklas Mattson[2], Lennart Minthon[2], Maria Nilsson[2], Rosita Nordkvist[2], Katarina Nägga[2], Camilla Orbjörn[2], Rik Ossenkoppele[2], Sebastian Palmqvist[2], Marie Persson[2], Alexander Santillo[2], Nicola Spotorno[2], Erik Stomrud[2], Håkan Toresson[2], Olof Strandberg[2], Michael Schöll[2], Ida Friberg[85], Per Johansson[85], Moa Wibom[85], Katarina Johansson[86], Emma Pettersson[86], Christin Karremo[86], Ruben Smith[86], Yulia Surova[86], Mattis Jalakas[87], Jimmy Lätt[88], Peter Mannfolk[88], Markus Nilsson[88], Freddy Ståhlberg[88], Pia Sundgren[88], Danielle van Westen[88], Ulf Andreasson[89], Kaj Blennow[89], Henrik Zetterberg[89], Lars-Olof Wahlund[90], Eric Westman[90], Joana Pereira[90], Jonas Jögi[91], Douglas Hägerström[91], Tomas Olsson[91] & Per Wollmer[91]

[85]Memory Clinic, Ängelholm Hospital, Skåne, Sweden. [86]Department of Neurology, Skåne University Hospital, Skåne, Sweden. [87]Department of Neurosurgery, Skåne University Hospital, Skåne, Sweden. [88]MRI Unit, Lund University, Lund, Sweden. [89]Clinical Neurochemistry Laboratory, University of Gothenburg, Gothenburg, Sweden. [90]MRI Unit, Karolinska Institutet, Solna, Sweden. [91]PET UNIT, LUND UNIVERSITY, Lund, Sweden.

