## [Peer Review File · Nature Communications]

Reviewers' comments:

Reviewer #1 (Remarks to the Author):

This manuscript describes the application of the ESM model of network-based spread of tau in the human brain, to large neuroimaging datasets containing tau PET scans. The ESM is an important advance and the present work is a natural extension and application of this approach. The results are also interesting with high correlations (model evidence).

Despite these strengths I find that the paper is lacking in many important ways, reducing my enthusiasm considerably. These are listed below.

1) Novelty. ESM has been an established approach, with several recent publications. It appears closely related to even earlier models that use network diffusion. Both approaches have now been reported numerous times on several datasets. A recent paper by Acosta et al on the ADNI study shows similar results, although it used ADNI regional atrophy rather than tau used here. I struggled to obtain a new insight from this manuscript that wasn't already contained in published literature, and could not come up with anything noteworthy. ESM was applied exactly as in published literature, with no new modeling aspects. To be fair, this wasn't the purpose of the paper anyway. Thus, despite very interesting results I do not find a meaningfully novel advance being reported here.

2) Methodology. Non-specific off target tau binding is a real problem in the field and the authors have presented an interesting approach of fitting bimodal Gaussian mixtures to remove them. Although I generally like the idea of some statistical approach to do this, I remain unconvinced that this is the right one. The presented approach has the key feature of removing subcortical regions (exactly the ones that cause "trouble"), but is couched in a statistical language that hides that fact. It seems that off target regions in the subcortex show a predominantly unimodal SUVR distribution, hence are cleanly removed by this approach. Fig 1 for instance does not show any of the regions we normally find to have off target binding (choroid plexus, thalamus, striatum). The authors are to be commended for proposing a principled statistical approach for this issue, but it would be far simpler and more forthright to simply remove these regions like many others have done before. It would also be necessary to report all results with and without this procedure, to assess whether the choice of off target removal is the key driving factor behind presented results.

3) Functional connectomes. I am puzzled by the use of functional connectomes (FC) in the context of tau spread. Tau can spread either through extracellular spaces or through neuronal projections. There is no plausible manner in which it can physically spread across functional connections, unless they have an underlying anatomic connection (which they are already getting from DTI tractography). How can tau spread to a remote location that does not actually connect to the current one? If it is through indirect hops, then ESM (and any network spread model) is already accounting for that.

In summary, this paper is well written, has interesting results and presents a validation of ESM on human tau data. But it has insufficient advance over the current state of the art, and has some important methodological issues that reduce enthusiasm.

Reviewer #2 (Remarks to the Author):

In this manuscript, the authors investigate the hypothesis that tau propagates via connectivity, either functional or structural connectivity, and that this process is facilitated by the presence of amyloid. To investigate this, they compared simulated epidemic spread models to data from 312 individuals from the ADNI database or the Swedish BioFinder Study. The authors find evidence for tau progression via connectivity, of which the effect sizes were stronger than those observed over Euclidian space (nearby spread). While this propagation pattern was irrespective of the amount of amyloid, regions with

elevated amyloid showed greater tau than predicted. The development of the tau-PET tracers has invigorated the interest in examining tau propagation and how this may interact with amyloid given that both proteinopathies show a typical topography in the brain. The idea in this paper is not entirely novel (e.g. (Franzmeier et al., 2019; Hoenig et al., 2018; Jacobs et al., 2018; Sepulcre et al., 2017)), but it adopts an interesting approach. The manuscript is well written. There are several issues that require attention, mainly on the analytical/methodological side and the lack of clinical relevance:

1. Sample: Participants came from two different cohorts. The authors did not describe how these cohorts may differ, may suffer from different selection biases or how they adjusted their models for these influences. Furthermore, of the MCI group, 64% was evaluated as amyloid positive. According to the recent research guidelines, these individuals may not have underlying AD pathologic change. Do the results change when excluding the MCI amyloid negative individuals? In addition, different methods for amyloid pathology were used: Florbetapir, Flutemetamol or CSF. Each of these methods have different signal-to-noise properties and also different sensitivities to diffuse versus fibrillar amyloid. How do the authors deal with these images?

2. PET methodology: It is unclear whether these PET images were corrected for partial volume effects? The Desikan-Killiany atlas does not contain 83 regions, but 64. In the methods the authors refer to 83 regions? While the GMM approach is indeed aimed at detecting different distributions in the data (which works very well for amyloid), the use of this method for tau is most likely more dependent on the exact composition of the sample and the sample size. Therefore, it would be valuable to compare and validate this method to the z-scoring methods based on (younger) individuals with "no" pathology (Cho et al., 2016; Grothe et al., 2017). In addition, I wonder how this approach deals with the off-target binding that influences the hippocampal signal. Choroid plexus signal is on average higher than hippocampal tau binding, but shows a similar distribution. Previous studies have regressed choroid plexus out of the hippocampal signal (Wang et al., 2016). Given that the hippocampus improved the model (also in the amyloid negative individuals), I wonder how much of this may be driven by off-target binding. In fact, off-target binding in subcortical regions, including the hippocampus, correlates strongly with age. Have the authors corrected their analyses for age (or sex, given the reported sex-differences in tau pathology)?

3. DTI analyses: How do these ADNI individuals differ from the cohort of the tau PET data? The authors used data of young healthy controls for the functional connectivity data, but healthy older and cognitively impaired individuals for the structural connectivity data. What was the rationale behind this choice and how may this difference in cohorts have influenced the results?

4. Results: on page 14 the authors report that using the bilateral entorhinal cortex epicenter explained 54.9% variance of the model of progression of tau. Have the authors examined possible left-right differences, given that other studies have reported asymmetry?

5. Clinical relevance: What I miss most in this manuscript is how these findings relate to the clinical symptoms in patients? The authors did not correlate their findings to cognitive performance, even though that data is available. The discussion also remains very much on the level of discussing the model fit. It would be great if these findings can be integrated and discussed with regard to their potential clinical relevance.

6. APOE: Given the high proportion of APOE carriers: have the authors investigated possible differences between carriers and non-carriers?

7. Discussion: there are several points that need clarification or further discussion:

a. The authors do not spend much time on discussing the regional patterns (role of amygdala?) and the fact that sometimes the correlation with the predicted result or the Braak staging is not great.

b. The authors should be more cautious in interpreting their data in a causal way. This is cross-

sectional data and any inference regarding progression or the role of amyloid is based on associations. The only way to infer causality is through interventions. Also, at several places in the discussion, the authors indicate that they have "strong" evidence for tau spreading, this should be toned down given the observational nature of the study.

c. Functional and structural connectivity are different processes: where structural connectivity refers to anatomy, functional connectivity represent communication between brain regions that may be mediated by third regions and may not follow the anatomical patterns. In terms of tau propagation, they may also reflect different mechanisms of cell-to-cell transfer. This deserves more thought and discussion.

d. Page 19: the authors consider low amyloid (below the threshold) as possibly being PART. PART is a neuropathological concept and is difficult to establish with amyloid PET, given the detection capabilities of amyloid PET (~ Thal stage 2 or 3). These interpretations should be reformulated.

e. It is unclear to me how the authors concluded based on this data that tau progression proceeds "slowly"?

8. Introduction: minor issues:

a. the authors state that tangle formation in the medial temporal lobe is part of normal aging. However, when tau reaches the hippocampus, this is usually associated with increases in amyloid pathology and increases in cognitive decline and thus may not be so innocuous.

b. This is not the first manuscript examining tau propagation along connections (e.g. (Franzmeier et al., 2019; Hoenig et al., 2018; Jacobs et al., 2018; Sepulcre et al., 2017). A clear discussion of the existing literature in the introduction or discussion is necessary. Furthermore, how these findings align or differ from previous studies or what they add to the existing knowledge should be discussed.

References mentioned in this review

Cho, H., Choi, J.Y., Hwang, M.S., Kim, Y.J., Lee, H.M., Lee, H.S., Lee, J.H., Ryu, Y.H., Lee, M.S., and Lyoo, C.H. (2016). In vivo cortical spreading pattern of tau and amyloid in the Alzheimer disease spectrum. *Ann Neurol* 80, 247-258.

Franzmeier, N., Rubinski, A., Neitzel, J., Kim, Y., Damm, A., Na, D.L., Kim, H.J., Lyoo, C.H.,

Cho, H., Finsterwalder, S., et al. (2019). Functional connectivity associated with tau levels in ageing, Alzheimer's, and small vessel disease. *Brain* 142, 1093-1107.

Grothe, M.J., Barthel, H., Sepulcre, J., Dyrba, M., Sabri, O., Teipel, S.J., and Alzheimer's Disease Neuroimaging, I. (2017). In vivo staging of regional amyloid deposition. *Neurology* 89, 2031-2038.

Hoenig, M.C., Bischof, G.N., Seemiller, J., Hammes, J., Kukolja, J., Onur, O.A., Jessen, F., Fließbach, K., Neumaier, B., Fink, G.R., et al. (2018). Networks of tau distribution in Alzheimer's disease. *Brain* 141, 568-581.

Jacobs, H.I.L., Hedden, T., Schultz, A.P., Sepulcre, J., Perea, R.D., Amariglio, R.E., Papp, K.V., Rentz, D.M., Sperling, R.A., and Johnson, K.A. (2018). Structural tract alterations predict downstream tau accumulation in amyloid-positive older individuals. *Nat Neurosci* 21, 424- 431.

Sepulcre, J., Sabuncu, M.R., Li, Q., El Fakhri, G., Sperling, R., and Johnson, K.A. (2017). Tau and amyloid-beta proteins distinctively associate to functional network changes in the aging brain. *Alzheimers Dement*.

Wang, L., Benzinger, T.L., Su, Y., Christensen, J., Friedrichsen, K., Aldea, P., McConathy, J., Cairns, N.J., Fagan, A.M., Morris, J.C., and Ances, B.M. (2016). Evaluation of Tau Imaging in Staging Alzheimer Disease and Revealing Interactions Between beta-Amyloid and Tauopathy. *JAMA neurology* 73, 1070-1077.

Dear Reviewers,

We are very appreciative for the opportunity to revise and resubmit our manuscript concerning simulating the spread of tau through macroscale connections in the human brain. The Reviewers offered a number of insightful comments and constructive suggestions. In enacting these suggestions, we feel we have produced a more complete manuscript and have boosted our confidence in our results. A detailed response to each Reviewer comment follows below.

In summary, many of the Reviewers' points related to methodological choices regarding the preparation of tau-PET data used as input for the ESM. These concerns included partial volume correction, confound regression, sampling choices, connectome choices, and regional normalization strategies, among others. We opted to look at the impact of each of these choices, including every possible combination, leading to the fitting of over 400 models. While some choices like partial volume correction and inclusion/exclusion of MCI- subjects made little difference, certain decisions had an important impact on the performance of our model. Exploring these methodological choices validated our mixture modeling approach as an appropriate normalization strategy, but also revealed that regression of confounding signals related to age and choroid plexus binding lead to improved model performance. A positive outcome of this analysis is that our best fitting models, using only connectivity information and diffusion properties, now explain 70% of the total variance in regional tau-PET signal in the brain—a significant improvement on pre-review model performance.

We additionally validate our findings over connectomes from two new datasets, further clarify our results in amyloid-negative individuals, and add many points to the discussion based on Reviewer suggestions. After addressing each comment, all of our main findings remain intact. However, thanks to creative suggestions by the Reviewers, we have added some novel aspects to the manuscript, including some interesting findings relating to individual hemispheric asymmetry in tau-PET distribution.

We hope the Reviewers will agree that the manuscript is now much improved, and we are once again thankful for the Reviewers spending their time and energy in lending their expertise to our study.

Reviewers' comments:

Reviewer #1 (Remarks to the Author):

This manuscript describes the application of the ESM model of network-based spread of tau in the human brain, to large neuroimaging datasets containing tau PET scans. The ESM is an important advance and the

present work is a natural extension and application of this approach. The results are also interesting with high correlations (model evidence).

Despite these strengths I find that the paper is lacking in many important ways, reducing my enthusiasm considerably. These are listed below.

1) Novelty. ESM has been an established approach, with several recent publications. It appears closely related to even earlier models that use network diffusion. Both approaches have now been reported numerous times on several datasets. A recent paper by Acosta et al on the ADNI study shows similar results, although it used ADNI regional atrophy rather than tau used here. I struggled to obtain a new insight from this manuscript that wasn't already contained in published literature, and could not come up with anything noteworthy. ESM was applied exactly as in published literature, with no new modeling aspects. To be fair, this wasn't the purpose of the paper anyway. Thus, despite very interesting results I do not find a meaningfully novel advance being reported here.

Response: We appreciate the Reviewer's perspective here, but would like to point out some important and novel aspects of the paper that perhaps the Reviewer is missing.

For starters, the reviewer refers to a paper (Acosta et al.) indicating the results are similar to our study. Previous applications of the network diffusion model in Alzheimer's disease have all used cortical thickness or volumetric analysis (e.g. Raj et al., 2012 Neuron; Torok et al., 2018 Brain). It is essential to understand that these measures are not equivalent to measuring paired-helical filament structures consisting of abnormally aggregated tau protein. To do so would assume that tau is the only cause of atrophy in the brain of AD patients, and we know this is not true. Particularly in LOAD (most of ADNI and of our BioFINDER cohort as well), multiple neuropathologies are at work and many have been directly associated with brain atrophy, including vascular disease, TDP-43, lewy body pathology. Using network diffusion on gray matter measure therefore conflates multiple pathologies that could each exert network diffusion behavior or not. Importantly, gray matter atrophy is also a downstream target for each of these pathologies, further complicating temporal dynamics, and tau is frequently seen in regions before any measureable neurodegeneration is observed (e.g. La Joie et al. In Press, Sci. Transl. Med.). This latter point is important because we are ultimately interested in the spread of tau itself, not the downstream consequences of tau spreading. Perhaps for these reasons, the predictions of models described in Acosta et al. actually demonstrate effect sizes ($r^2 = 0.24-0.31$) substantially lower than our original models ($r^2 = 0.55 - 0.62$), as well as those presented in the revision (r^2 up to 0.70).

The advantage of our approach is that we are using a tracer that autoradiography studies (e.g. Marquie et al., 2015,2017; Lowe et al., 2016) have confirmed binds

directly to paired helical filament tau pathology. Herein lies the novelty of our approach – previous diffusion modeling studies have tried to use gray matter measures as a proxy for tau with limited success because large enough samples of tau-PET data were not available. By measuring tau pathology directly in a large sample as we do here, we are more confident that our signal is not confounded by other pathologies. No previous studies we are aware of have applied diffusion modeling to tau-PET data, and this may explain why the effect sizes of our models are substantially higher than previous studies. We also go further to provide information regarding how amyloid interacts with our models and, due to Reviewer suggestions, have added novel findings related to asymmetric distribution of tau pathology.

A final point to be made is one already made by the Reviewer – the purpose of the manuscript is not to introduce a new advance to the ESM, it is to apply the ESM to a large and valuable dataset to test a biological hypothesis. However, we do introduce an approach for normalizing tau-PET data that may end up being a useful technological advance (see response to the next Comment below).

Altogether, we respectfully disagree with the Reviewer with respect to novelty of our manuscript. In summary, we apply a diffusion model directly to tau-PET data for the first time, leading to much stronger results than in previous studies, adding information about interaction with amyloid, finding novel results with respect to amyloid negative individuals and hemispheric asymmetric spreading, and introducing a new approach for normalizing tau-PET data.

See also our response to Reviewer 2, Comment #8b (minor issue), which is somewhat related. We have also addressed some the Reviewer's concerns in a new paragraph in the discussion (page 16).

2) Methodology. Non-specific off target tau binding is a real problem in the field and the authors have presented an interesting approach of fitting bimodal Gaussian mixtures to remove them. Although I generally like the idea of some statistical approach to do this, I remain unconvinced that this is the right one. The presented approach has the key feature of removing subcortical regions (exactly the ones that cause "trouble"), but is couched in a statistical language that hides that fact. It seems that off target regions in the subcortex show a predominantly unimodal SUVR distribution, hence are cleanly removed by this approach. Fig 1 for instance does not show any of the regions we normally find to have off target binding (choroid plexus, thalamus, striatum). The authors are to be commended for proposing a principled statistical approach for this issue, but it would be far simpler and more forthright to simply remove these regions like many others have done before. It would also be necessary to report all results with and without this procedure, to

assess whether the choice of off target removal is the key driving factor behind presented results.

Response: *The Reviewer brings up some interesting points. However, it was not our intention to find a methodological approach that removes “problem” regions. In fact, since specific subnuclei of some subcortical structures do show tau pathology at autopsy (e.g. Aggleton et al. 2016, Brain), we were hoping our approach could detect this subtle pathology. If we did, we would have been happy to include these structures in our analysis. The purpose of this approach was to identify regions that do not show abnormal tau-PET signal, and remove these regions. We generally feel that using an empirical approach based on biological assumptions is a superior option to arbitrarily removing regions that tend to be problematic.*

However, in the end, the Reviewer’s suspicion proved correct. In response to many of Reviewer 2’s requests, we reran many different models with different types of input data and preprocessing methods. The ROIs selected by the mixture models changed depending on the input data, and often ended up selecting some subcortical regions. Likely for this reason, model performance tended to be much better on average when we simply removed all subcortical regions, rather than only removing regions selected by the mixture modeling approach, or removing no regions (see Figure below, where error bars represent variation in model fit across different connectomes, preprocessing strategies, PVC, and sampling strategies – see responses to Reviewer 2).

As a result, we move forward with the Reviewer’s request to simply remove subcortical regions from the model. However, our mixture-modeling approach still proved to be a critical preprocessing step. We ran our models using two other reference strategies: i) regional normalization of SUVR values along a 0-1 scale (equivalent to simply using SUVR values as input—ESM requires values to be between 0 and 1); ii) the reference strategy described in the original ESM paper (Iturria-Medina 2014, PLoS Comp. Biol.). As can be seen below, the mixture modeling strategy improves ESM fit substantially compared to these other reference strategies, irrespective of the underlying connectome data. The

confidence intervals show the SEM across model r^2 when varying PVC, preprocessing, and sampling strategies (i.e MCI- included or not; see responses to Reviewer 2).

We expect that a larger dataset, future techniques for data-cleaning, or a more sophisticated mixture modeling approach may eventually achieve our mutual goal of statistically identifying and removing problematic regions without bias.

Due to comments from both Reviewers, the Methods section has been substantially rewritten, and the comparisons described here have been addressed in Methods section 6.4, Results section (2.2) and Figure S2.

3) Functional connectomes. I am puzzled by the use of functional connectomes (FC) in the context of tau spread. Tau can spread either through extracellular spaces or through neuronal projections. There is no plausible manner in which it can physically spread across functional connections, unless they have an underlying anatomic connection (which they are already getting from DTI tractography). How can tau spread to a remote location that does not actually connect to the current one? If it is through indirect hops, then ESM (and any network spread model) is already accounting for that.

Response: The choice to use rsfMRI in addition to DTI was twofold.

The first reason has nothing to do with biology -- we wanted to show that the results were consistent no matter how we were measuring regional connectivity. Both DTI and fMRI are imperfect measurements for different reasons. fMRI as a technique may be methodologically further advanced due to global harmonization efforts, such that many solutions and workarounds have been developed to deal with some of its specific limitations e.g. motion. In contrast, mainstream DTI tractography pipelines still contend with key issues, for example, the gyral bias, curving and crossing fibers, or failure to detect smaller or unmyelinated tracts

(Jbabdi et al., 2015 Nat Rev Neurosci). Some might argue that, because of these factors, well-processed fMRI connectivity may actually be more reliable than DTI connectivity at estimating regional macroscale connectivity. We remain neutral in this debate, but feel it is important to reproduce findings across multiple measurements, seeing that none of them are perfect.

Second, the Reviewer is correct that tau cannot physically spread “indirectly”. However, some hypothesis of tau spread have proposed that it is not physical oligomers that are spreading, but that instead, tau is the result of a specific biological state that can be initiated through some other mechanism(s). In such cases, direct anatomical connection would technically not be necessary for communication of this pathological state. By using fMRI, we are accounting for this possible hypothesis of tau spread by allowing regions that activate together to be connected irrespective of direct connections.

In general, we found that connectome choice resulted in small changes to overall model fit (see Figure below), with an advantage of DTI tractography. However, importantly, model success did not depend on the modality used to generate the underlying connectome.

Due to this concern being voiced by both Reviewers (see Reviewer 2, comment #3 and 7c), we have mentioned this connectome comparison in a new paragraph in the Discussion, reproduced here for convenience:

“Tau can be directly secreted into extracellular space, and mechanisms have been described for subsequent cellular uptake (c.f. Fuster-Matanzo et al., 2018), leading to the hypothesis that tau may be propagated to neighboring neurons. This idea is not supported by our data, where neuronal connectivity patterns provided a better description of the *in vivo* spatial distribution of tau. Another hypothesis stems from the observation that tau has an excitatory effect on neurons (DeVos et al., 2013), but is also secreted by activated neurons (DeVos et al., 2013, Pooler et al., 2013). These two observations have lead to the

idea of an excitotoxic cascade, where the presence of tau excites neurons, leading to overstimulation of connected neurons, which in turn leads to secretion of tau, and so forth. This latter hypothesis cannot be ruled out based on our data, as it is still predicated on the spreading of pathological events across communicating neurons. In our study, we fit the ESM over two different measures of macroscale connectivity, and the choice of modality comes with different sets of assumptions and limitations. DTI tractography endeavors to directly measure white matter connections between brain regions, and may therefore be the most appropriate choice, but also suffers from important methodological limitations such as the gyral bias (Jbabdi et al, 2015). On the other hand, rsfMRI connectomes are conflated by indirect connectivity (Jbabdi et al, 2015; e.g. Fig S2D), which do not fit with the hypothesis of direct axonal spread. However, one can imagine a scenario where a region may act as way station for tau propagation without itself expressing pathological tau due to (say) its genomic environment. Additionally, alternative hypotheses of tau propagation involving network propagation of a pathological (e.g. excitotoxic or tau overproduction) state would not necessarily require direct connections. In our data, DTI tractography-based connectomes consistently showed superior model fit compared to models fit over other connectomes (Fig 2, S2D), once again lending support to the cell-to-cell transmission hypotheses, though model fit was ultimately high and reproducible across both connectivity modalities. Next-generation tractography may provide improved models in the future (Maier-Hein et al., 2017), but both measures of connectivity appear to be sufficient for fairly high performing simulations of tau spread.”

In summary, this paper is well written, has interesting results and presents a validation of ESM on human tau data. But it has insufficient advance over the current state of the art, and has some important methodological issues that reduce enthusiasm.

Response: We thank the Reviewer for his or her opinion. We have addressed the issue of novelty in previous comments, and we also feel that an approximate doubling of effect sizes does represent an advance over the state of the art. However, we are not sure which methodological issues the Reviewer has concerns with, as none of the above Reviewer comments speak to technical issues that might impact the rigor or meaning of our results. Please see also our responses to Reviewer 2, which address many other aspects of our approach, and lead to even stronger models.

Reviewer #2 (Remarks to the Author):

In this manuscript, the authors investigate the hypothesis that tau propagates via connectivity, either functional or structural connectivity, and that this process is facilitated by the presence of amyloid. To investigate this, they compared simulated epidemic spread models to data from 312 individuals from the ADNI database or the Swedish BioFinder Study. The authors find evidence for tau progression via connectivity, of

which the effect sizes were stronger than those observed over Euclidian space (nearby spread). While this propagation pattern was irrespective of the amount of amyloid, regions with elevated amyloid showed greater tau that predicted. The development of the tau-PET tracers has invigorated the interest in examining tau propagation and how this may interact with amyloid given that both proteinopathies show a typical topography in the brain. The idea in this paper is not entirely novel (e.g. (Franzmeier et al., 2019; Hoenig et al., 2018; Jacobs et al., 2018; Sepulcre et al., 2017), but it adopts an interesting approach. The manuscript is well written. There are several issues that require attention, mainly on the analytical/methodological side and the lack of clinical relevance:

1. Sample: Participants came from two different cohorts. The authors did not describe how these cohorts may differ, may suffer from different selection biases or how they adjusted their models for these influences. Furthermore, of the MCI group, 64% was evaluated as amyloid positive. According to the recent research guidelines, these individuals may not have underlying AD pathologic change. Do the results change when excluding the MCI amyloid negative individuals? In addition, different methods for amyloid pathology were used: Florbetapir, Flutemetamol or CSF. Each of these methods have different signal-to-noise properties and also different sensitivities to diffuse versus fibrillar amyloid. How do the authors deal with these images?

Response: The Reviewer raises some important points. The Reviewer is correct that there are important demographic differences (Education level) and differences in selection (greater proportion of amyloid positive subjects in BioFINDER), and these changes are now detailed in Table S1. With respect to our analyses with tau, we were not very concerned with cross-cohort differences. For starters, both cohorts used the same radiotracer (AV1451/flortaucipir) and were processed in-house with the same pipeline. Furthermore, the ESM is fit within-subject, not across groups – only the epicenter is chosen based off the entire dataset. However, to ensure the model is working in both cohorts, we summarized results separately for ADNI and BioFINDER subjects using the best fitting model (see Response to Comment #2 below).

Overall, the model achieves high accuracy for both cohorts, indicating the model is robust across datasets of varying compositions. However, differences in the overall tau load across cohort are visible, and model performance is better in ADNI. To investigate whether the observed differences could be attributed to cohort differences in demographics or in overall cortical tau load, we used Nearest Neighbors algorithms to subsample the BioFinder cohort so that it best matched ADNI with respect to a) demographics (Age, Sex, Education, ApoE4 genotype) and b) tau (mean tau probability across all ROIs), respectively. Once we applied our matching algorithm for a), the new BioFINDER and ADNI exhibited no differences in sex, or ApoE status, though the BioFINDER cohort was still older and less educated ($p < 0.05$). For b), the matched BioFINDER sample showed no difference in cortical tau load to ADNI. For both a) and b), we tested overall model fit of the two new matched BioFINDER samples compared to model fit in the ADNI sample (see Figure below).

Matching the BioFinder sample to ADNI based on demographics partially improved fit in the BioFinder sample. Conversely, matching BioFinder to ADNI based on overall tau load resulted in a model fit almost identical to ADNI. These results suggest that the difference in model fit seen between ADNI and

BioFINDER may be attributable to a greater overall tau load in BioFINDER. Overall, these results suggest that the ESM can explain tau spreading patterns well in disparate samples, and differences in fit across cohorts can be explained by differences in tau load between cohorts, rather than some unknown cohort effects. Given that we've shown amyloid exerts regional effects on model performance, it is possible that poorer model performance in cohorts with more tau may be of function of greater amyloid deposition. Interestingly, while the ADNI and BioFINDER samples differed in amyloid positivity ($p < 0.001$), the BioFINDER sample matched to ADNI in tau load did not ($p = 0.36$), further supporting the notion that amyloid may be systematically affecting our model.

We have added these results to the manuscript, Section 2.2 and Figure S3.

Next, the Reviewer is correct that a number of MCI individuals are included that do not have amyloid and who are therefore unlikely to experience cognitive impairment due to Alzheimer's disease, and unlikely to show much tau burden outside the medial temporal lobe. This is also true of our amyloid negative controls – we do not expect them to express much tau, if any. The ESM fitting is unaffected by a lack of tau expression, as the ratio between the beta (production) and delta (clearance) parameters should preclude much expression of tau outside the epicenter (see Fig 1B). The reason for inclusion is to increase the sample N, which should improve the sensitivity of our regional mixture model analysis, as well as to increase the number of amyloid-negative individuals for contrast. As can be seen below across different connectomes, the model is mostly unchanged when removing amyloid-negative MCI subjects. Error bars below represent model fit variation across preprocessing decisions and PVC strategies (see response to Reviewer 2, Comment #2 below). It is notable, however, that the best fitting overall model did not include these MCI- subjects (see Fig S2A, response to Reviewer 2, Comment #2 below).

Finally, the Reviewer is correct that the amyloid tracer is different across the two cohorts, and that this could conceivably impact classification of subjects as positive or negative. A number of studies has shown high concordance in classification between florbetapir, flutemetamol and CSF abeta40-42 (e.g. Landau et al., 2014 EJNMMI; Palmqvist et al., 2015 Neurology, Mattsson et al., 2014 Ann. Clin. Translat. Neurol, etc). However, to emphasize this point, we show that ESM performance is still high in subjects negative for both CSF and PET, who are additionally ApoE4-negative and cognitively normal.

We do agree, however, that mixing amyloid images taken from two different tracers may not be advisable. Therefore, we instead created the amyloid map used for Figure 6 from a large (partially) separate dataset using only one tracer (florbetapir). The results were extremely similar to our previous results, as can be seen below in the new Figure 6, and were strengthened (perhaps by the increased sample sizes adding stability to the amyloid measures).

2. PET methodology: It is unclear whether these PET images were corrected for partial volume effects? The Desikan-Killiany atlas does not contain 83 regions, but 64. In the methods the authors refer to 83 regions? While the GMM approach is indeed aimed at detecting different distributions in the data (which works very well for amyloid), the use of this method for tau is most likely more dependent on the exact composition of the sample and the sample size. Therefore, it would be valuable to compare and validate this method to the z-scoring methods based on (younger) individuals with “no” pathology (Cho et al., 2016; Grothe et al., 2017). In addition, I wonder how this approach deals with the off-target binding that influences the hippocampal signal. Choroid plexus signal is on average higher than hippocampal tau binding, but shows a similar distribution. Previous studies have regressed choroid plexus out of the hippocampal signal (Wang et al., 2016). Given that the hippocampus improved the model (also in the amyloid negative individuals), I wonder how much of this may be driven by off-target binding. In fact, off-target binding in subcortical regions, including the hippocampus, correlates

strongly with age. Have the authors corrected their analyses for age (or sex, given the reported sex-differences in tau pathology)?

Response: *The Reviewer makes several excellent points, which we will respond to piecemeal. First, with respect to the Desikan-Killiany (DKT) atlas, the Reviewer is correct that the atlas has 64 cortical regions. If you count the 14 subcortical regions and the 5 cerebellar regions, the total amount is 83. These specific regions were generated from the Freesurfer parcellations of each subject, which include several other additional regions. We have changed the wording to be clearer about the regions included.*

“Pre-processing of PET data resulted in mean regional tau-PET SUVR values from the freesurfer-derived Desikan-Killiany-Tourville (DKT) atlas (Desikan et al., 2006) extracted from each individual's native space PET image. Only cortical and subcortical regions overlapping with the MindBoggle DKT atlas (Klein & Tourville, 2012) were included, leaving 78 regions in total”

The Reviewer's next point concerns our choice of data normalization. We agree with the Reviewer that the mixture modeling approach used here is somewhat sensitive to sample size and composition, as already mentioned in the Methods (Section 6.4) and Limitations (Page 17). However, using our mixture model procedure on the data substantially outperforms model fit in comparison to two other types of data: i) regional normalization of SUVR values along a 0-1 scale (equivalent to simply using SUVR values as input—ESM requires values to be between 0 and 1); ii) using the reference strategy described in the original ESM paper (Iturria-Medina 2014, PloS Comp. Biol.). As can be seen below, the mixture modeling strategy improves ESM fit substantially compared to these other reference strategies, irrespective of the underlying connectome data. The confidence intervals show the SEM across model r^2 when varying PVC (see below), preprocessing (see below) and sampling strategies (i.e MCI- included or not, see previous comment).

The Reviewer continues by suggesting a comparison to a reference strategy involving normalization by younger, amyloid negative individuals. We disagree with this approach for many reasons. The most important is that AV1451 scans between young and old individuals differ substantially in off-target binding (Baker et al., 2019 JNM) in ways that are unrelated to tau pathology. However cognitively intact, amyloid negative elderly subjects also do not make a good “reference” group, because this group may show age-associated tau that may be very relevant to Alzheimer’s disease (Crary et al., 2014 Acta Neuropathol). We nonetheless perform a regional W-score approach (La Joie et al., 2012 J Neurosci) using cognitively normal, amyloid negative individuals as a reference group, and refit the ESM model. The Reviewer additionally asks about the possible impact of age, sex, and choroid plexus signal on our results. We therefore test all of these associations using two approaches: 1) the aforementioned W-scoring, which normalizes regional SUVR values by the amyloid-negative cognitively normal elderly adjusting for age, sex and choroid plexus binding; 2) regressing age, sex and choroid plexus binding out of each region across all subjects. Finally, we performed each of these preprocessing strategies using each of the input data variations from the previous figures, and across each connectome. The comparisons of model fit across all of these conditions is shown below, where the error bars represent variation in model performance based on different PVC (see below) and sampling strategies (see response to previous comment).

The mixture-modeling approach for defining tau probabilities resulted in the best model fit across all preprocessing strategies. Using a “healthy” control group to normalize probability values (W-score approach) as the Reviewer suggested did not appear to enhance model fit across any data input conditions. On the other hand, regressing out age, sex and choroid plexus signal actually enhanced model fit for models using the mixture-model approach for input data, but not the other data input conditions. This latter finding is interesting, as regressing out age and choroid plexus signal is expected to result in a cleaner tau signal (Baker et

al. 2019 JNM; Lee et al., 2018 JAD). The fact that this preprocessing step enhances models with mixture-model input data, but seems to actually decrease model fit for other data input strategies, may once again suggest that the mixture model approach is better representing pathological signal.

Finally, the Reviewer mentions partial volume correction (PVC). We did not initially perform PVC, but we can appreciate the possibility that PVC could improve the quality of the PET data, and therefore could improve our models. However, after processing our data using the popular geometric transfer matrix (GTM; Rousset et al. 1998) PVC strategy, we did not find that PVC substantially changed the results. For each boxplot below, each dot represents a different model, varying across reference strategy, preprocessing strategy and connectome. The line between boxes connecting the dots shows how PVC changes model r^2 when all other parameters remain stable. The rightmost plot shows only models using the (more successful) mixture modeling strategy for input data. As can be seen, PVC has a variable affect on model fit, but it does not improve the top models, nor does it improve ESM performance on average.

Altogether, the mixture modeling strategy for generating tau probabilities outperforms other data input strategies across all conditions, and is enhanced by first regressing out age, sex and choroid plexus signal from the regional data. Simply using normalized SUVR values results in models with modest fits ($r^2 \sim 0.25$) that are inferior to the mixture model strategies and are not aided by regression. The Iturria-Medina 2014 method does not appear to be appropriate normalization strategy for this data. Generally, W -scoring using a “normative” sample leads to worse models and PVC does not change any of the results. Finally, all of these results were consistent when looking across only PiB negative individuals, as shown below:

These results and others above and below have been added the Methods (6.3, 6.4, 6.5) and Results (Section 4.2), and summarized in Figure S2. Model fit statistics for every combination of conditions has been added to Table S2, and parameters of the top 25 models can be found in Figure S2A. In addition, this procedure has allowed us to now identify the best model across conditions. Using the regression approach suggested by the Reviewer, a DTI connectome based on young subjects (see next comment), and excluding MCI- subjects (see previous comment), we are able to improve model fit of the best model to an r^2 of 0.7, and an average within-subject r^2 of 0.52. Global fit of this best fitting model is shown below, and is now represented in Figure 3:

3. DTI analyses: How do these ADNI individuals differ from the cohort of the tau PET data? The authors used data of young healthy controls for the functional connectivity data, but healthy older and cognitively impaired individuals for the structural connectivity data. What was the rationale behind this choice and how may this difference in cohorts have influenced the results?

Response: *Demographic information for the non-overlapping group of ADNI subjects with DTI data has been added to Supplementary Table 1. As can be seen, there are some significant demographic differences between the cohorts: DTI sample CN were older, MCI had more ApoE4 carriers, and had generally more amyloid positive subjects than the ADNI tau sample; DTI sample was generally more educated and had less amyloid than BioFinder). We however do not anticipate these demographic differences will have a large influence on the ESM; as can be seen above and below, the results can be reproduced across connectomes using different modalities and very different samples.*

The Reviewer brings up a good point about the apparent asymmetry between modality and age group across connectomes. There was no particular rationale behind this decision, it was based purely on data availability at the time of writing the manuscript. We have since preprocessed fMRI data from a non-overlapping cohort of 189 ADNI subjects using the same NIAK pipeline as was used to process the young cohort from COBRE. Demographics of these subjects have been added to Supplementary Table 1. In addition, we have processed DTI tractography data from a sample of 60 young subjects from the CMU-60 cohort using an identical processing pipeline as was used to process our ADNI data. We now have fMRI- and DTI-based connectomes from samples of both older/impaired and young adults. Looking at the figures in response to the previous Reviewer comment, one can appreciate that model fit does not vary substantially across different connectomes, though structural connectomes appear to afford an advantage when the data has been normalized with mixture modeling. This is summarized in the figure below, where error bars represent variability in model fit across variation in preprocessing, PVC and subsampling (i.e. MCI- removed or not) strategies:

4. Results: on page 14 the authors report that using the bilateral entorhinal cortex epicenter explained 54.9% variance of the model of progression of tau. Have the authors examined possible left-right differences, given that other studies have reported asymmetry?

Response: This is a very interesting point. We had not looked into left-right asymmetry, but at the Reviewer’s request, we have fit the ESM using only the left or right entorhinal cortex alone as the epicenter, and compared this to models using both hemispheres as an epicenter. The results are shown below, separately for each connectome. Error bars represent variation in model fit across varying preprocessing, PVC and subsampling strategies (see above comments).

We observed expected differences between connectomes. When using functional connectomes, choice of left-right intercept does not matter. This is expected given the high degree of within-region heterotopic connectivity (and indirect connectivity in general) observed in rsfMRI data. In other words, due to the high functional connectivity between left and right entorhinal cortex, choosing

one or the other is nearly equivalent to choosing both. However, because the left and right entorhinal cortices do not share direct anatomical connections (using our tractographic measures), using only one hemisphere as an epicenter substantially reduced model fit. We also did not observe strong evidence that model fit across the whole sample improved by using one hemisphere over another. This suggests that tau does not start preferentially in the left or right entorhinal cortex across the population.

However, the above results do not preclude the possibility that tau may appear preferentially in one hemisphere for individual subjects. To approach this question, we ascertained the best fitting model epicenter for each subject individually (results in Figure below). We found that 59.3% of subjects data were best fit using a right limbic epicenter, 31.1% a left limbic epicenter, and only 9.6% an extra-limbic epicenter (where limbic referred to the entorhinal cortex, hippocampus, parahippocampus, or amygdala; panel B). Interestingly, right-limbic epicenters were extremely prevalent in CN- subjects (~80%), but decreased with disease progression, while prevalence of left-limbic and other epicenters increased (panel C). We also found that epicenter hemisphere was significantly ($p < 0.001$) associated with total tau asymmetry, and that this effect was present across the disease spectrum and became stronger in later disease stages ($ps < 0.01$; panel E). In addition, a left-limbic epicenter was associated with greater left temporo-parietal tau, but less right frontal tau, after FDR correction and adjustment for age, sex and disease status (panel F). This may point to differing cortical expression of tau depending on the hemisphere of origin. Epicenter hemisphere was not associated with global tau load. Regarding demographic difference between individuals with different hemispheres, we saw a general trend of individuals with right-hemisphere limbic epicenters being younger, adjusting for disease status ($p = 0.01$; panel D). We did not however see an impact of epicenter hemisphere on sex, education, ApoE4 status, or cognition (MMSE or CDR) when adjusting for disease status.

Taken together, this data supports the notion of individual differences in hemispheric asymmetry of tau progression, and this hemispheric preference may be age associated and related to the cortical expression of tau. However, the finding that model fit across the entire sample is improved by using a bilateral epicenter suggests that, even in cases of hemispheric asymmetry, tau propagation occurs through both hemispheres with a fairly close temporal proximity. These results have now been added as a new section to the results (Section 2.5) and our visualized in Figure 7.

5. Clinical relevance: What I miss most in this manuscript is how these findings relate to the clinical symptoms in patients? The authors did not correlate their findings to cognitive performance, even though that data is available. The discussion also remains very much on the level of discussing the model fit. It would be great if these findings can be integrated and discussed with regard to their potential clinical relevance.

Response: The focus of this manuscript is whether the pattern of tau spreading observed in the brains of people with Alzheimer's disease is informed by network topology. While tau is highly correlated with cognition, there is substantial variation in the extent of tau propagation that can occur before clinical cognitive impairment or dementia manifest (e.g. Hoenig et al., 2017 Neurobiol Aging). Therefore, we are interested in testing hypotheses of tau spread that are independent of cognitive status. To this point, and as can be seen in the Figure below (now part of Figure S4), the ESM explains the spread of tau fairly well across different disease states.

While our focus is explaining the biological process of tau spread, there are still clinical applications to our work, and we agree with the Reviewer that we can do a better job of outlining these applications in the Discussion. For example, knowledge that tau propagation can be predicted by connectivity patterns can help clinical trials seeking to track tau spread over the course of a trial, by giving a probability of where tau will spread given a baseline scan. In addition, if tau is spreading through cells and if we know neuronal activity promotes tau secretion, this may point toward treatments that seek to lower the overall excitatory tone of the brain, which could attenuate the velocity of tau propagation. Ultimately, the

data presented here are testing a mechanistic hypothesis rather than a clinical question, and this is why we focus on how well this hypothetical model fits the data, and what can be improved from the perspective of disease biology. Nevertheless, we have now added some points to discussion concerning the clinical relevance of our findings.

“In addition, the ESM has potential as a clinical tool by estimating where tau will spread based on individual regional patterns. Knowledge of the expected pattern of tau spread will be helpful in designing regional outcome measures in future treatment trials directed against tau aggregation.”

The Reviewer requests that we “correlate [our] findings to cognitive performance”. There have already been many papers correlating tau levels to cognition (e.g. Scholl et al., 2016; Cho et al., 2017; Benjanin et al., 2017; Mattsson et al., 2019, Pontecorvo et al., 2019), including in a mixed ADNI and BioFINDER (Vogel et al., 2019 Hum Brain Mapp). Therefore, we do not feel adding correlations between tau and cognition fits the direction or purpose of the manuscript. We are testing how well a model of tau spreading fits the actual observed spread of tau, independently of cognitive status. However, the Reviewer can perhaps appreciate that many aspects of our model results corroborate expectations with regard to cognitive status. For example, predicted tau probabilities increase with worse disease status (Figures above and below).

6. APOE: Given the high proportion of APOE carriers: have the authors investigated possible differences between carriers and non-carriers?

Response: At the Reviewer’s request, we examined whether model fit differed between ApoE4 carriers and non-carriers. We used linear models to examine the effect of ApoE4 status on within-subject model fit, and permutation tests (5000

permutations) were used to assess significant differences in global pattern fit between carriers and non-carriers. These tests were initially conducted using the “best-fitting” model (see end of response to Reviewer Comment #2). We did not find an effect of ApoE4 status on within-subject model fit across the whole sample, nor within different diagnostic groups (CN-, CN+, MCI+, AD+). We also did not find an effect of ApoE4 status on global model fit across the whole sample, though a trend emerged where model fit was slightly better in MCI subjects carrying an ApoE4 allele compared to MCI non-carriers ($p=0.068$). To further explore this, we repeated the same permutation test across the other three connectomes (young rsfMRI, old rsfMRI, old DTI). The trend was present when using the young ($p=0.068$) and old ($p=0.069$) rsfMRI connectomes, and a similar effect emerged in ApoE4 carrying AD individuals as well ($p=0.028$) using the old rsfMRI connectomes only (see Figure below; * $p<0.05$; x $p<0.1$).

The MCI ApoE4 carriers and non-carriers did not differ in total tau nor in amyloid load (recall that all subjects were amyloid-positive). In addition, no regional differences in tau signal were seen between MCI ApoE4 carriers and non-carriers after adjustment for multiple comparisons. Below, we have plotted the global model fit across MCI ApoE4 carriers and non-carriers. Interestingly, the mean absolute errors of these two fits are nearly identical.

Taken together, the effects we have seen above were of a low magnitude and inconsistent. Therefore, we not feel there is enough evidence to conclude differing model performance depending on ApoE4 status. Due to the equivocal nature of these results, we have opted not to add these results into the paper.

7. Discussion: there are several points that need clarification or further discussion:

a. The authors do not spend much time on discussing the regional patterns

(role of amygdala?) and the fact that sometimes the correlation with the predicted result or the Braak staging is not great.

Response: We thank the Reviewer for this suggestion. We had not formally examined the correspondence of our model (nor the observed data) to the Braak stages, and now took the opportunity to do so, visualized in the Figure below using stages described in Scholl et al., 2016 (now Figure 2).

We had already mentioned in the manuscript that the model performed worse further along in the disease progression (Section 2.2, Discussion page 13), and had dedicated an entire paragraph to discussing possible reasons for this (page 15). However, we have now added the above figure to the main text (Figure 2), and we are now more explicit about the correspondence of our model's predictions to Braak staging:

“While our model recapitulated the early stages of tau spreading accurately (Braak I-III), later stages (IV-VI) were modeled less accurately, with a systematic underestimation of tau in regions prone to early and high-volume beta-amyloid aggregation.”

As for the amygdala, it is a known early hotspot for tau aggregation. Its prominence in our data may be more related to its small size. In contrast, while the tau reaches the anterior hippocampus already by Braak stage II, it only propagates the length of the hippocampus in later stages (Lace et al., 2009

Brain). *While we are interested in this point, we regret that we will not have space to discuss the specific role in the amygdala.*

b. The authors should be more cautious in interpreting their data in a causal way. This is cross-sectional data and any inference regarding progression or the role of amyloid is based on associations. The only way to infer causality is through interventions. Also, at several places in the discussion, the authors indicate that they have “strong” evidence for tau spreading, this should be toned down given the observational nature of the study.

Response: We agree with the Reviewer’s point here, and have dialed back the language with respect to our findings.

c. Functional and structural connectivity are different processes: where structural connectivity refers to anatomy, functional connectivity represent communication between brain regions that may be mediated by third regions and may not follow the anatomical patterns. In terms of tau propagation, they may also reflect different mechanisms of cell-to-cell transfer. This deserves more thought and discussion.

Response: This comment converges strongly with Reviewer 1 Comment #3. Our initial motivation for using both functional and structural connections was to validate that our model is working across different measures of macroscale brain connectivity. However, the Reviewer brings up a number of important points about how these measurements differ. We have now added an additional paragraph to the Discussion section discussing how the choice of connectome influences assumptions of our hypothetical model, and how the limitations of each modality may influence model performance. The paragraph is printed below:

“Tau can be directly secreted into extracellular space, and mechanisms have been described for subsequent cellular uptake (c.f. Fuster-Matanzo et al., 2018), leading to the hypothesis that tau may be propagated to neighboring neurons. This idea is not supported by our data, where neuronal connectivity patterns provided a better description of the *in vivo* spatial distribution of tau. Another hypothesis stems from the observation that tau has an excitatory effect on neurons (DeVos et al., 2013), but is also secreted by activated neurons (DeVos et al., 2013, Pooler et al., 2013). These two observations have lead to the idea of an excitotoxic cascade, where the presence of tau excites neurons, leading to overstimulation of connected neurons, which in turn leads to secretion of tau, and so forth. This latter hypothesis cannot be ruled out based on our data, as it is still predicated on the spreading of pathological events across communicating neurons. In our study, we fit the ESM over two different measures of macroscale connectivity, and the choice of modality comes with different sets of assumptions and limitations. DTI tractography endeavors to directly measure white matter connections between brain regions, and may therefore be the most appropriate choice, but also suffers from important methodological limitations

such as the gyral bias (Jbabdi et al, 2015). On the other hand, rsfMRI connectomes are conflated by indirect connectivity (Jbabdi et al, 2015; e.g. Fig S2D), which do not fit with the hypothesis of direct axonal spread. However, one can imagine a scenario where a region may act as way station for tau propagation without itself expressing pathological tau due to (say) its genomic environment. Additionally, alternative hypotheses of tau propagation involving network propagation of a pathological (e.g. excitotoxic or tau overproduction) state would not necessarily require direct connections. In our data, DTI tractography-based connectomes consistently showed superior model fit compared to models fit over other connectomes (Fig 2, S2D), once again lending support to the cell-to-cell transmission hypotheses, though model fit was ultimately high and reproducible across both connectivity modalities. Next-generation tractography may provide improved models in the future (Maier-Hein et al., 2017), but both measures of connectivity appear to be sufficient for fairly high performing simulations of tau spread.”

d. Page 19: the authors consider low amyloid (below the threshold) as possibly being PART. PART is a neuropathological concept and is difficult to establish with amyloid PET, given the detection capabilities of amyloid PET (~ Thal stage 2 or 3). These interpretations should be reformulated.

Response: We appreciate this interesting point by the Reviewer. Neurofibrillary pathology in cases with sparse neuritic plaques does still fall into the definition of PART (Crary et al., 2014), but we understand the Reviewer’s concern that some of the effects observed in our amyloid-negative individuals might be driven by subthreshold amyloid. To address this, we isolated a group of 64 cognitively normal subjects with negative amyloid-PET scans, without marginally decreased CSF abeta40-42, and without any copies of an ApoE4 allele. We hope the Reviewer would agree that these subjects would be the least likely to demonstrate sub-threshold amyloid. However, model fit in this group was still quite high despite very low levels of tau in most regions. Furthermore, despite these low levels of tau, the overall pattern still followed a Braak stage-like progression (see Figure below, panel B, right). The figure below also highlights four such individuals from this “triple negative” group, each demonstrating good model fit and a Braak like progression despite low tau levels (panel C). In addition, model fit was generally high across the majority of our amyloid-negative subjects (panel A), and based on our knowledge, it would not be likely for such a high proportion of subjects to demonstrate undetectable amyloid. In contrast, tau in the entorhinal cortex, hippocampus and other medial temporal lobe structures is common in normal aging (Braak et al., 2006, 2015; Crary et al., 2014).

We agree with the Reviewer that PART has been heretofore challenging to establish using tau-PET. However, other groups have published findings describing low-level tau signal in amyloid-negative individuals, and this signal has been occasionally associated with memory measures as well (e.g. Maass et al., 2018). In our manuscript, we corroborate well-known findings that tau-PET binding in amyloid-negative healthy controls is indeed low and subthreshold by previous standards of measurement. However, we also show that the regional pattern of this subthreshold binding is nonetheless predictable using entorhinal cortex connectivity, and indeed falls into a similar pattern as amyloid-positive subjects. We are therefore observing a pattern of tau-PET binding that is very low, and yet that takes on a particular pattern consistent with that seen in AD. While we cannot conclude that what we are seeing is PART, we do not feel it is an unreasonable speculation to include in the discussion of our findings. Thus, we have opted to keep this comment in the manuscript, though we have now edited sections of the discussion in order to make clear that this is only speculation, and that there are other possibilities. For example, see excerpt below:

“The inability of amyloid-PET to identify sparse amyloid burden in the medial temporal lobes (Jack et al., 2013) may lead to the possibility that undetectable levels of amyloid pathology may be driving the observed relationships.”

e. It is unclear to me how the authors concluded based on this data that tau progression proceeds “slowly”?

Response: We agree with the Reviewer that this conclusion is somewhat speculative. We know that tau accumulates faster in amyloid-positive compared to amyloid-negative subjects, and that increased baseline tau is associated with increased speed of accumulation (e.g. Pontecorvo, 2019 Brain). In this sense, tau accumulation will likely be slow in low-tau amyloid negative individuals, if

accumulation occurs at all. Our data suggest a progressively higher AV1451 signal in regions more connected to the entorhinal cortex, in amyloid-negative individuals, possibly indicative of a progression from the entorhinal cortex in these subjects. This is what lead us to the hypothesis that tau is indeed spreading in these subjects, albeit at a lower rate. However, given the speculative nature of these conclusions, we have opted to simply remove the word “slowly” from the sentence.

8. Introduction: minor issues:

a. the authors state that tangle formation in the medial temporal lobe is part of normal aging. However, when tau reaches the hippocampus, this is usually associated with increases in amyloid pathology and increases in cognitive decline and thus may not be so innocuous.

Response: As the Reviewer states, tau in the hippocampus is related to increased amyloid pathology and cognitive decline (Lace et al., 2009 Brain; Robinson et al., 2011 Brain), but these relationships are associations. At the individual level, tau is frequently observed in the hippocampus in absence of cognitive decline and amyloid plaque deposition (e.g. Crary et al., 2014 Acta Neuropathol). Indeed, Braak Stage IV has been observed at autopsy in amyloid-negative individuals (Crary et al.), and the majority of 60-69 year olds have NFT stage I-II (Braak & Del Tredici 2015, Brain). While early Braak stage tau is unlikely to be harmless, it is fairly well accepted to be a part of normative aging.

To appease the Reviewer, we have slightly changed the wording of this sentence:

“However, tau tangle aggregation specifically in the medial temporal lobes is a common feature of normative aging.”

b. This is not the first manuscript examining tau propagation along connections (e.g. (Franzmeier et al., 2019; Hoenig et al., 2018; Jacobs et al., 2018; Sepulcre et al., 2017). A clear discussion of the existing literature in the introduction or discussion is necessary. Furthermore, how these findings align or differ from previous studies or what they add to the existing knowledge should be discussed.

***Response:** This comment converges somewhat with Reviewer 1, Comment #1. We have updated the introduction to include the Hoenig, Jacobs papers (the Franzmeier paper was already mentioned). The Sepulcre paper is now mentioned in the Discussion. We agree with the Reviewer that a paragraph in the discussion comparing our approach and findings to the existing tau-spreading literature would be very useful. While we addressed this to some extent in the introduction, we have expanded upon it in a new paragraph in the Discussion (excerpt below). In summary, with regard to our approach, other studies have not examined tau propagation along connections as the Reviewer states; rather they*

have observed covariance between connectivity measures and tau measures. Our study is unique in its modeling of an actual propagation scheme, starting at a single point, and progressing through secondary and tertiary seeding events over time. This is likely why the effect sizes of our findings are substantially larger than those currently reported (e.g. Franzmeier, Acosta, etc).

“The results of the ESM represent an advance on previous human studies testing the spreading hypothesis of tau. Many previous studies addressing this hypothesis have elected to examine covariance between tau patterns and brain networks, usually measured with rsfMRI. Jones et al. and Hoenig et al., described overlap between data-driven tau-PET covariance networks and resting-state functional networks. Franzmeier et al. and Ossenkoppele et al. each went further to show correlations between rsfMRI connectivity and cross-subject covariance in tau-PET signal, within networks or across the whole brain. Sepulcre et al. instead used longitudinal tau covariance across spatially distributed regions to infer connectivity between those regions. Each of these studies represent clues that tau spreading and connectivity are related in humans. However, they do not construct, test or simulate models of tau spreading. The ESM simulates the spread of tau from the entorhinal cortex through a cascade of secondary seeding events informed by macroscale functional or structural connections, a process that is designed to mimic the hypothetical spreading of tau. This model can explain upwards of 70% of the spatial variation of tau in the human brain, representing a substantial improvement over the aforementioned associational studies, as well as over studies using similar diffusion models on structural MRI measures (e.g. Torok, Acosta). While our simulation explains the tau-PET data to an unprecedented degree, it is imperfect and remains indirect evidence of tau spreading. However, it also provides a first step toward a tau spreading simulation model, which can be improved, perturbed and applied in numerous contexts.”

Reviewers' comments:

Reviewer #1 (Remarks to the Author):

This revision has addressed several critical points from the first round, and has consequently improved. I appreciate the attention to reviewer concerns, and the new explanation for why functional connectomes were used.

Regarding specific comments, I want to highlight only the key one regarding novelty. As the authors have correctly noted, the "novelty" lies in applying the ESM method on a new dataset, one that uses tau instead of atrophy. This is not at variance to my original comment. I agree whole heartedly with the value of tau imaging in comparison to atrophy, and also agree that the presented results support the ability of ESM to recapitulate empirical data. However my point remains, that this manuscript is an exercise in validating an existing model on new, better data. To that extent it is a useful addition to the field. There is not however a new modeling approach or a new insight that was not previously known. It is therefore up to the editors to decide whether this represents the level of advance that merits publication in their journal.

Reviewer #2 (Remarks to the Author):

The authors did a tremendous amount of work to examine the influence of different preprocessing steps and other confounders on their findings. In addition, the manuscript has a much better flow, making it much easier to understand what was done and what it means.

- An important test of the model presented here that is still missing would be to show that the ESM approach does not perform well when taking another region, for example, lateral occipital, as seed epicenter. This would strengthen the conclusions as well as the methodology.
- Figure 2: can you indicate the proportion of MCI and AD subjects that were staged in each Braak Stage.
- At some point in the manuscript, this authors should acknowledge that the best fitting statistical does not equate that is the model that closest to the real underlying biology. In particular, the fact that the best-fitting model did not use PV correction, might indicate that atrophy (and possibly other processes) may contribute to the result.
- On page 6, the authors write: "The epidemic spreading model was particularly effective in predicting the early progression of tau, but diverged more from the observed tau pattern over time". Can they explain this in more detail? Which regions were discordant with the predicted probabilities? Figure 3 in the amyloid+ plots, shows that in the first, second and fifth row there are two regions with large residuals (far from the regression line): which regions are these?
- Figure 5B: in these plots, Braak stage II (hippocampus) shows consistently a large difference between predicted and observed tau probability. But stage I and stage III show a good fit. How do the authors explain this?
- Figure 6: Do regions within the underestimated composite have a higher magnitude of tau burden than those in the overestimated composite? If so, then an alternative explanation could be that these regions have more amyloid, because of a parallel tau-process. It is not necessarily interactive.
- Page 14, line 262: I don't understand why MTL amyloid is being referenced here? In general, diffuse amyloid is harder to detect with Florbetapir

Minor comments:

- Page 2, line 9: remove "if any"
- Page 2, line 11: "However, tau tangle aggregation specifically in the medial temporal lobes is a common feature of normative aging". The authors should nuance this statement as recent work by Lowe et al., (2019) indicated that entorhinal tau is associated with lower memory performance in amyloid+ and amyloid- individuals
- Page 12 line 200: what do the authors mean with "issues relating to the quantitative measurement

of tau"? A mixture modelling approach has also quantitative issues (biased by the sample in which it is used). Please rephrase this sentence.

- Page 12, line 213: "leads to" : causal effects were not shown in this paper and the findings should not be presented as such

Dear Reviewer #2,

We are thankful for the opportunity to revise our manuscript, and we are once again appreciative of your excellent suggestions. In accordance with your comments, we have added what we feel is an important finding, showing that the entorhinal cortex represents the best model epicenter across all brain regions in the DKT atlas. We also show that regional amyloid contributes significantly to model estimation even when controlling for local tau pathology. These additions, along with those detailed below, have once again improved the manuscript and increased our confidence in the results. We are grateful for your close attention and continued improvement of our manuscript.

Specific responses follow:

Reviewer #2 (Remarks to the Author):

The authors did a tremendous amount of work to examine the influence of different preprocessing steps and other confounders on their findings. In addition, the manuscript has a much better flow, making it much easier to understand what was done and what it means.

- An important test of the model presented here that is still missing would be to show that the ESM approach does not perform well when taking another region, for example, lateral occipital, as seed epicenter. This would strengthen the conclusions as well as the methodology.

Response: The Reviewer brings up an excellent point. We reran the ESM substituting each left-right pair of ROIs (33 pairs in total) as the model epicenters, and measured model fit. The results are visualized in the Figure below, where blue bars represent model fit across the whole sample, and red bars represent average within-subject model fit.

As can be seen, the entorhinal cortex demonstrated the best overall model fit, which is consistent with our expectations and strengthens our confidence in our methodological approach. As one might expect, other MTL regions made for decent epicenters, while “control region” epicenters (e.g. primary sensory) resulted in very poor model fitting. This Figure has been added to the Main Text Figure 3.

- Figure 2: can you indicate the proportion of MCI and AD subjects that were staged in each Braak Stage.

Response: We apologize for the miscommunication on our part. We did not divide individuals into Braak stages at any point in this manuscript. However, we did divide **brain regions** into Braak stage ROIs, a la Schöll et al. 2016 Neuron. For example, if abnormal tau is first observed in Region A at Braak stage IV (according to Braak staging regime), Region A would be included in the Braak stage IV ROI. This paradigm is also used in Figure 5. Figure 2 instead is representing how well the distribution of raw tau-PET SUVR values, tau-positive probabilities and predicted tau probabilities across the sample conform to the expected distribution as described by Braak. As such, each individual has a

value for each Braak stage ROI. We have amended the Figure 2 caption to describe more clearly what is being shown:

“Tau-positive probabilities recapitulate Braak staging. Each brain region was divided into one of six "Braak stage" ROIs, based on which Braak stage abnormal tau is first observed in the region (as described in Schöll et al., 2016) (Left) Each row is a subject sorted top-bottom by least to most overall tau. Each column is an Braak stage ROI, sorted left to right by most to least overall tau. Warmer colors represent higher SUVR values (top), observed tau-positive probabilities (middle) or predicted tau-positive probabilities from the best-fitting ESM (bottom). (Right) The same relationship shown in a bar chart format. Conversion to tau-positive probabilities creates a sparse distribution of values demonstrating a progression reminiscent of the staging described in the autopsy literature.”

- At some point in the manuscript, this authors should acknowledge that the best fitting statistical does not equate that is the model that closest to the real underlying biology. In particular, the fact that the best-fitting model did not use PV correction, might indicate that atrophy (and possibly other processes) may contribute to the result.

Response: We whole-heartedly agree with the Reviewer's point that the best-fitting model is not necessarily the model that best approximates biology. We have therefore added the following statement to the Limitations:

“Finally, we tested the ESM over a number of different pre-processing decisions, and mostly describe results of best-fitting models. It is important to note that a model that best fits our data does not necessarily equate to a model that best fits biology. However, many different pre-processing combinations produced high-performing models (Supplementary Fig S2A), so we are confident that our results are not dependent on our pre-processing decisions”

While we generally agree with the Reviewer's comment, we do not feel that our PVC results are a good example of this. PVC is an imperfect technique that can add noise to the data. There is no consensus that PVC improves tau-PET signal, and it made little difference in our high performing models one way or the other.

- On page 6, the authors write:” The epidemic spreading model was particularly effective in predicting the early progression of tau, but diverged more from the observed tau pattern over time”. Can they explain this in more detail? Which regions were discordant with the predicted probabilities? Figure 3 in the amyloid+ plots, shows that in the first, second and fifth row there are two regions with large residuals (far from the regression line): which regions are these?

Response: The statement is based on the observation that larger residuals are observed on the left side of each of these graphs, indicating regions that have less tau (and are therefore further along in the progression) show greater discrepancy between predicted and observed. We have now quantified this explicitly in the Figure below. The figure quantifies the average residual (i.e. difference between predicted and observed tau) for regions falling into each Braak stage composite.

As can be seen, the general trend is increasingly worse fit with increasing Braak stage (with the exception of the hippocampus, which is poorly fit in all circumstances). This trend fits our observation that the model is less accurate further along in the progression. However, part of this can be explained by amyloid, as depicted in Figure 6. The above Figure has now been included in the manuscript as Supplementary Figure S5. Regarding the hippocampus, see our response to the next Reviewer comment.

The two regions far from the regression line the Reviewer refers to are the left and right entorhinal cortex, which appear to be underestimated using a few different connectomes, specifically in amyloid-positive subjects. However, as can be seen in the above Figure, the entorhinal cortex fitting is still better than those in later Braak stages in our best fitting model.

- Figure 5B: in these plots, Braak stage II (hippocampus) shows consistently a large difference between predicted and observed tau probability. But stage I and stage III show a good fit. How do the authors explain this?

Response: As we state in the discussion, even if the ESM fit is very good, it is far from perfect. The hippocampus appears to be underestimated in the amyloid-negative individuals – in other words, the tau probabilities are higher than the ESM predicts in these subjects. Given frequent reports of AV1451 signal in the hippocampus being difficult to manage (e.g. Cho et al., 2016; Ikonovic et al., 2017; Leuzy et al., 2019), it is unsurprising to see this region behaving somewhat

abnormally, even after our attempts to normalize it. However, importantly, both the observed and predicted data place the hippocampus as a region with early, non-zero tau signal, as one of the first 4-5 regions to show abnormal tau. Therefore, we do not feel the discrepancy the Reviewer has noted is prominent enough to cause alarm.

- Figure 6: Do regions within the underestimated composite have a higher magnitude of tau burden than those in the overestimated composite? If so, then an alternative explanation could be that these regions have more amyloid, because of a parallel tau-process. It is not necessarily interactive.

Response: The Reviewer brings up an interesting point. As the Reviewer predicted, underestimated regions have a higher magnitude of tau burden as well. However, in a model including both regional amyloid and regional tau, both terms remain significant. While this does not rule out the Reviewer's alternative explanation, it does suggest that the local impact of amyloid on model fit is independent of local tau.

We have added this finding to Results section 2.5, and an accompanying sentence at the end of Methods section 6.6.

- Page 14, line 262: I don't understand why MTL amyloid is being referenced here? In general, diffuse amyloid is harder to detect with Florbetapir

Response: We have amended this line to be more clear, removing reference to MTL amyloid:

“The inability of A-beta-PET to identify sparse A-beta burden, especially in cases with predominant diffuse plaques, may lead to the possibility that undetectable levels of A-beta pathology may be driving the observed relationships. However, we demonstrated an early Braak-like pattern of tau in individuals at very low likelihood of having A-beta pathology (cognitively normal, ApoE4-negative, CSF A-beta negative).”

Minor comments:

- Page 2, line 9: remove “if any”

Response: These words have been removed.

- Page 2, line 11: “However, tau tangle aggregation specifically in the medial temporal lobes is a common feature of normative aging”. The authors should nuance this statement as recent work by Lowe et al., (2019)

indicated that entorhinal tau is associated with lower memory performance in amyloid+ and amyloid- individuals

Response: We agree with the Reviewer that this is an important point, and it has already been made in the Discussion. However, at the Reviewer's request, we have changed the sentence to read:

"However, tau tangle aggregation specifically in the medial temporal lobes is a common feature of normative aging (Crary et al., 2014; Braak et al., 2015; Harrison et al., 2018), itself associated with subtle cognitive effects (Maass et al., 2018; Lowe et al., 2019)"

- Page 12 line 200: what do the authors mean with "issues relating to the quantitative measurement of tau"? A mixture modelling approach has also quantitative issues (biased by the sample in which it is used). Please rephrase this sentence.

Response: The sentence the Reviewer refers to limitations of tau-PET studies. We feel quantitative measurement of tau is a prominent limitation of these studies, and we chose an approach that attempts an alternative method. We are not saying our approach is without its own limitations – in fact we include a sentence about it in our Limitations section. However, the approach we have taken is in direct response to issues observed in previous studies. Therefore, we do not feel this sentence has any issues and we are choosing to keep it as is.

- Page 12, line 213: "leads to" : causal effects were not shown in this paper and the findings should not be presented as such

Response: We changed the "leads to" to "is associated with".

REVIEWERS' COMMENTS:

Reviewer #2 (Remarks to the Author):

The authors have done a great job in working through all my comments. I have no further comments and congratulate them with this manuscript.